



# Seismic imaging of dyke swarms within the Sorgenfrei Tornquist Zone (Sweden) and implications for thermal energy storage

Alireza Malehmir[1,*], Bo Bergman[2,3], Benjamin Andersson[4], Robert Sturk[4] and Mattis Johansson[2,3]

[1] Dept. of Earth Sciences, Uppsala University, Sweden
[2] Sweco Environment AB, Malmö, Sweden
[3] WSP (presently), Malmö, Sweden
[4] Skanska Sverige AB, Malmö, Sweden

*Correspondence to*: Alireza Malehmir (alireza.malehmir@geo.uu.se)

**Abstract.** There is a large interest and demand for green-type energy storage in Sweden for both short- and long-terms
(hours, days, weeks and seasons). While there are a number of approaches proposed (e.g., compressed air, geothermal and
thermal), only a few have commercially been demonstrated through up-scaling projects. Among these, the thermal energy
storage (TES) that stores energy (excess heat or cold) in fluids is particularly appealing. The excess energy then can be
stored underground in excavated caverns and used for large district heating and cooling purposes. For an up-scaling
underground thermal-energy-storage project within the Tornquist suture zone of Scania in southwest of Sweden three high-
resolution, each approximately one kilometer long, 5 m receiver and source spacing, seismic profiles were acquired.
Geologically, the site sits within the southern margin of the Romeleåsen Fault Zone in the Sorgenfrei Tornquist Zone (STZ)
where dolerite dyke swarms of Carboniferous-Permian age are observed striking in SE-NW direction for over 100s of
kilometers both on land and in offshore seismic and magnetic data (from Scania to Midland Valley in the UK). These dykes,
5-20 m thick, in the nearby quarries (within both Precambrian gneiss and quartzite) express themselves mostly sub-vertical.
They can therefore act as a good water/fluid barrier, which can be an important geological factor for any thermal energy
storage site. In the data acquisition, combined cabled- and wireless recorders were used to provide continuity on both sides
of a major road running in the middle of the study area. Bedrock depressions are clearly depicted in the tomograms
suggesting the possibility of weakness zones, highly fractured and/or weathered, in the bedrock and confirmed at several
places by followed-up boreholes. Several steeply dipping (60-65°) reflections were imaged down to 400 m depth and based
on a series of arguments are judged to be from dolerite dykes. This includes their orientations, strong amplitudes, regular
occurrences, and correlation with downhole logging data. In addition groundwater flow measurements within the
unconsolidated sediments and in bedrock suggests steeply-dipping structures are the dominant factor in directing water
towards SE-NW direction, which is consistent with the strike of the dyke swarm within the STZ. To provide further insight
into the origin of the reflections even the historical crustal-scale offshore BABEL lines (A-AA-AB) were revisited. Clear
multi-phase faults are observed as well as a Moho step across the Tornquist zone. Overall, we favour that the reflections are
of dolerite origin and their dip component (i.e., not sub-vertical) may imply a Precambrian basement (and dykes) tilting,
block rotation, towards NE as a result of the Romeleåsen reverse faulting. In terms of thermal storage, these dykes then may





be encountered during the excavation of the site and can complicate underground water flow should they be used as a fluid barrier in case of a leakage.

# 1 Introduction

Demand and chase for renewable-energy is rapidly increasing worldwide although with some temporal fluctuations in order to reduce $CO_2$ emissions and also to provide a sustainable, cheaper and alternative source of energy to the market (e.g., International Energy Agency, 2018; Alva et al., 2018; Ingleso-Lotz and Dogan, 2018). In Sweden, through a number of initiatives and cutting-edge technologies, a number of companies aim to upscale their ideas to provide large-district heating and cooling systems for both short- and long-term purposes. For example, Skanska Company has invented (Skanska Sverige AB, 2013) the idea of underground energy storage based on the mechanism that thermos keep hot and cold water for a long period of time (thermal energy storage or TES). This builds on the physical principle of "hot-water floating on cold water" (Håkansson 2016) allowing excess energy to be collected and used later; or simply for balancing energy consumption between daytime and nighttime. The concept however requires field demonstration, and large-scale underground developments (Skanska Sverige AB, 2013) and facilities where a good control and suitable subsurface geology is present. Thermal energy is stored in a fluid (e.g., water) in tanks or an excavated cavern. The cavern then is insulated by suitable geology (impermeable rocks) or though concrete walls. Since different fluids can be considered for this purpose (e.g., fluids used in low temperature systems), leakage and accidents must be avoided or leaked fluids are safely and fast removed in case of an accident.

To support such a development in the subsurface (at a depth of about 300 m) we conducted a seismic survey near the town of Dalby-Lund within the Scania Tornquist suture zone (Fig. 1) in southwest of Sweden during August 2015. The site is situated close to the RFZ (Romeleåsen fault/thrust and flexure zone) with a complex geologic and tectonic history (Erlström and Sivhed, 2001). Near-vertical dykes are observed from several quarries in the area crosscutting granitic-gneissic-amphibiotic rocks and form clear magnetic lineaments in the regional data (Fig. 1). These dykes likely have also acted as surfaces on which further faulting (until present day) have occurred. They are doleritic (diabase) in composition, Permian in age (Torsvik et al., 2008) and can be followed through the Scania region of Sweden and on offshore seismic data reported from the southern North Sea (Heeremans et al., 2004; Philips et al., 2017 and references therein).

The seismic survey had an initial objective of identifying depth to bedrock and if major bedrock undulations could be related to zones of weaknesses (fractured or/and altered) in bedrock. Nevertheless, given the high-fold seismic data acquired and rich reflectivity observed in the raw shot gathers, reflection data processing complemented the refraction data analysis. In this work, we show a series of northeast-steeply-dipping reflections that surrounds the planned underground storage and speculate on their origins to be of dolerite dykes. Furthermore, we revisited the historical offshore BABEL lines A-AA-AB (BABEL Working Group, 1990 and 1993; Fig. 1) in an attempt to check if these dykes were also observable. As a result, we

provide better images of a necking (keel or step) zone at the Moho level across the Tornquist zone that was proposed by Mazur et al. (2015) as a result of the collision between two Precambrian terranes with different crustal thicknesses.

**Figure 1: (a) Aeromagnetic anomaly and (b) bouguer gravity maps of the Scania region of Sweden showing the location of the study area south of the town of Dalby. Magnetic lineaments clearly mark the location of Permian dyke swarms within the Sorgenfrei Tornquist Zone (STZ) that is a horst structure within an active strike-slip tectonic setting. Inset map shows interpreted location of the dyke swarms of the same generation within the STZ. Dyke orientations and ages are based on Phillips et al. (2017 and references therein). In this work historical offshore BABEL lines A-AA-AB (green lines in the inset map) have also been revisited and results discussed. Aeromagnetic and gravity data were kindly provided by the Geological Survey of Sweden.**

## 2 Geology of the Sorgenfrei Tornquist Zone (STZ)

Trans-European suture zone or Tornquist zone in general is a major crustal scale boundary in Europe that separates the Precambrian East European Craton from the Phanerozoic orogens of South Western Europe. In the Scania (Skåne) of Sweden and northern Denmark it is a major strike-slip feature (Fig. 1), reactivated many times through its history, tectonic inversion, and referred as the Sorgenfrei Tornquist Zone (STZ) while in northern Poland it is referred as the Teisseyre



Tornquist Zone (TTZ). It is traditionally thought to be the boundary of the lithospheric plate of Baltica that existed between late Neoproterozoic and early Paleozoic (Cocks and Torsvik, 2005). A crustal keel approximately 20-km wide associated with a Moho-step has been inferred based on a major gravity low across the zone in Poland by Mazur et al. (2015); see also Thybo (2000), Grad et al. (2002), Malinowski et al. (2005), Guterch and Grad (2006). Offshore seismic data suggest that the

Tornquist zone extends to the southern Norwegian North Sea (Pegrum, 1984). Based on the results from the BABEL offshore seismic data, a number of authors have proposed that the Tornquist zone is a listric fault under the Hanö Bay Basin (e.g., Blundell, 1992; Erlström et al., 1997; Thybo, 2000; Meissner and Krawczyk, 1999). According to Meissner et al. (2002) during the Late Cretaceous, tectonic inversion of a rigid upper mantle led to a large-scale compressive tectonics. Within the Tornquist zone, this led to a series of reverse faults in the upper crust as a result of transferring stress in the lower

ductile crust undergoing buckling to the upper crust (see also Sopher et al., 2016 and references therein). Hanö Bay Basin (see section of BABEL A-AA in Fig. 1) formed as a result of syn-inversion extension (pull-apart basin due to fault curvature or an extension) associated with significant strike-slip faulting in the Tornquist zone (Blundell, 1992).

Structurally, the most striking geological feature in the Skåne area is the Romeleåsen horst that stretches over 30 km along the STZ. It was formed by fault reactivations in the Tornquist zone during Mesozoic and Tertiary compression (Bergerat et

al., 2007). The town of Dalby is situated on the southwestern margin of the Romeleåsen reverse fault system. Together with the Vomb-Fyledalen fault system (reverse), the area between these fault systems is uplifted (Romele Ridge) and is the main reason for the gravity high (basement high) observed in the region (Fig. 1b).

Lithologically in the Skåne area, rocks are primarily composed of Proterozoic gneiss-amphibolite (nearly 50%-50%) to sedimentary rocks of various ages. In the Dalby quarry north of the study area, amphibolites dip often towards southwest as

lens-shaped lenticular bodies (Fig. 2a). Evidence of faulting and hydrothermal activities is evident in the quarry. A dominant feature however is the Permo-Carboniferous doleritic (mafic) dykes crosscutting gneissic-amphibolitic rocks (Fig. 2a) and sometimes quartzitic (metamorphosed bedded sandstone) rocks in the area (Fig. 2b). They occur in a regular order often 5-20 m thick generally striking in NW-SE direction as evident in the magnetic data over the Skåne area (Fig. 1a). While often these dykes are considered to be sub-vertical, sometimes in field photos from nearby quarries they appear to be steeply

dipping (e.g., Bergerat et al., 2007). Phillips et al. (2017) observe the same generations of the dykes (320-270 Ma) in the southern Norwegian North Sea albeit as dipping reflections (35-50° north dipping) and argue for the role of tectonic inversion (faulting) and basin-basement flexure (block rotation) for their current geometry. Although a matter of a debate, Phillips et al. (2017) claim the first dyke swarm images observed in reflection seismic data in the world. They also suggest the dyke swarm is part of a major magmatic system that has a radial shape extending for over 800 km connecting to those

observed in the Midland Valley dyke suit (302-292 Ma) in the UK.





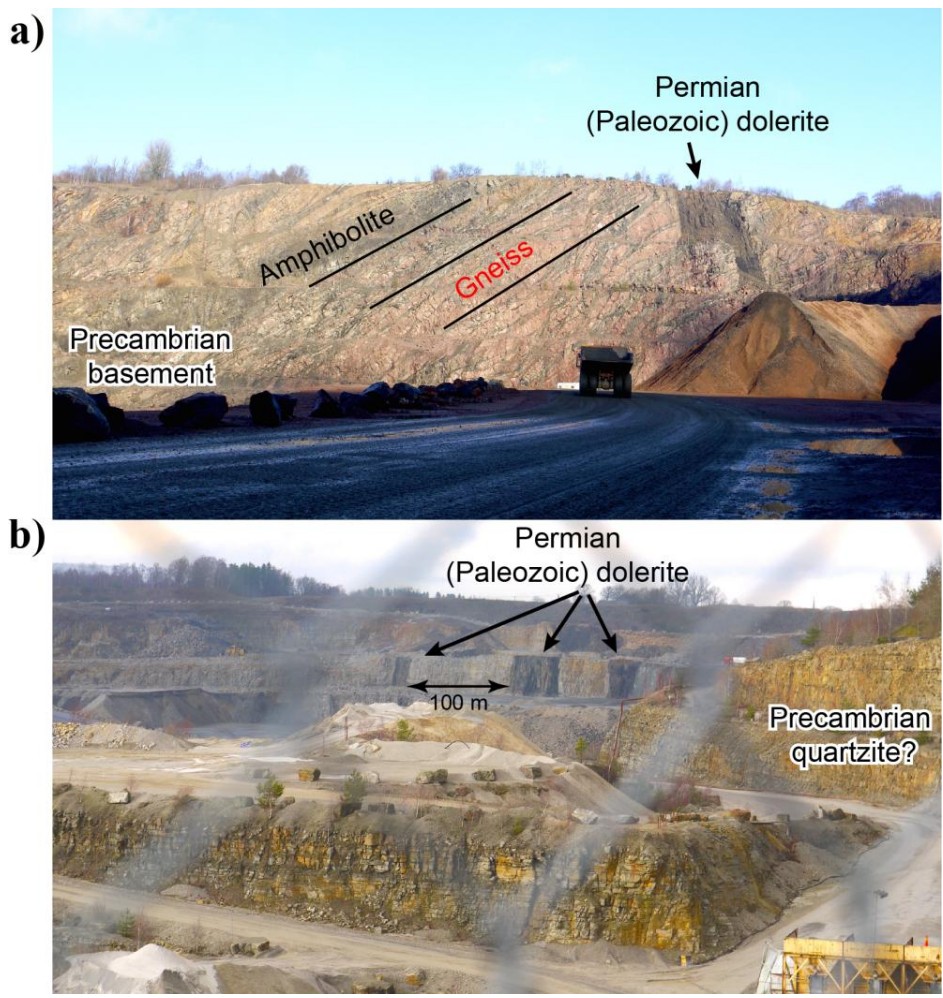

**Figure 2: Example photos showing typical doleritic dykes within the Scania area (a) from the Dalby rock quarry (about 500 m north of the study area), and (b) a quartzite quarry (about 2 km northwest of the study area). The dykes appear, 10-50 m thick, in a regular order typically 50-100 m far apart and crosscut stratigraphy and amphibolites. They tend to be sub-vertical in most places that they have been observed on land. In the Dalby quarry these dykes make up to 5% of the rocks excavated. Gneiss and amphibolite are abundant and typically equality constitutes the remaining bulk volume of the excavated rocks. Photos from Alireza Malehmir (July 2015).**

## 3 Seismic survey

### 3.1 Seismic data acquisition

Three high-resolution, 5 m shot and receiver spacing using 141-172 receivers, refraction and reflection seismic profiles were acquired during five days; 2 days for profile 4, 1.5 day for profile 2 and 1.5 day for profile 3 (Fig. 3). Profile 3 was not in the



original plan but decided to be acquired at the site because of its position and being parallel to profile 2 where geological structures are favorable in this direction than perpendicular to it. Some delays with the seismic source and noise due to wind caused problem when acquiring the data along profile 4. A Bobcat-mounted drophammer (500 kg) was used to generate the seismic signal (e.g., Place et al., 2015). To provide continuity from one side of the road to another, 51 wireless recorders

connected to 10-Hz geophones and operating in an autonomous mode were used (Fig. 3). In total approximately 2.6 km long reflection seismic data were acquired.

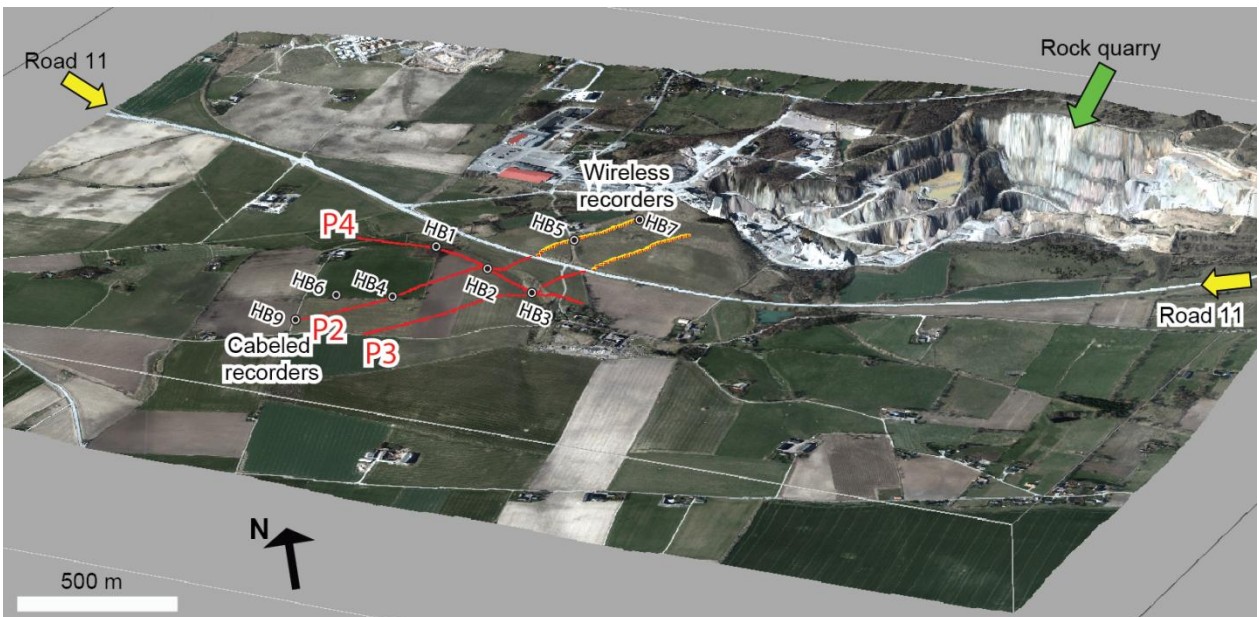

**Figure 3: Locations of the seismic profiles (P2, P3, and P4), in Dalby southern Sweden, draped onto the high-resolution LiDAR elevation surface of the study area. A combination of cabled- and wireless-recorders was used for the data acquisition to allow**

**long-offset recording and to overcome issues related to the high-speed road 11. Black filled-in circles are the drilled boreholes (after the seismic data acquisition) in the study area.**

GPS times of the source impacts (micro-second accuracy) recorded on the cabled sensors (also 10-Hz geophones) were used to extract the data from the wireless recorders and then these were merged together. Three shot records per source position

were generated and vertically stacked to improve signal-to-noise ratio. Noise from the high-speed road 11 and cars passing was at times significant but the vertical stacking of the repeated shots helped to partly cancel this noise. At occasions due to the proximity to the Malmö airport, airplane noise was also significant and led to delays in the data acquisition. Table 1 summarizes some of the main acquisition parameters used in this study.

Figure 4 shows a collection of photos taken during the seismic survey and the equipment used. Although road 11 runs in the

middle of the study area, the seismic data show excellent quality particularly along profiles 2 and 3, and slightly noisy, due to high wind, along profile 4.



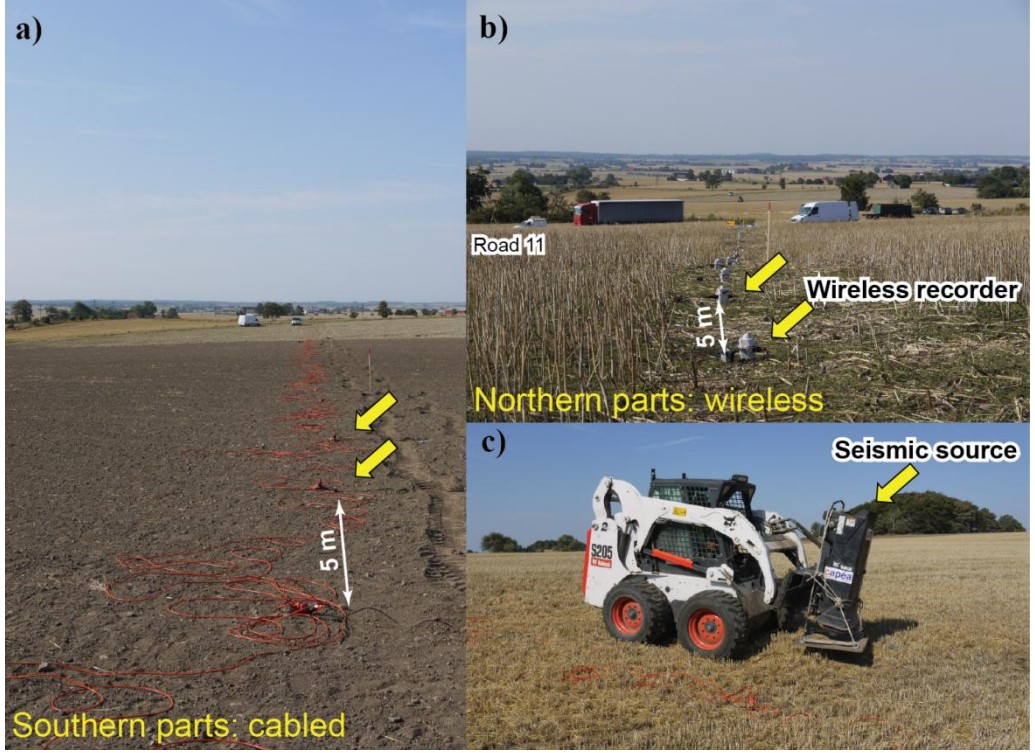

**Figure 4: Field photos showing how the seismic data were acquired using a combination of (a) cabled- and (b) wireless-recorders (connected to 10-Hz geophones). (c) A 500-kg Bobcat-mounted drophammer was used as seismic source. Only at a few locations close to the road 11 and on eastern side of profile 4 when farming was not finished shooting was not possible.**

Figures 5a and 6a show example shot gathers after vertical stacking of the three repeated shot records (3 hits at every shot location) from profiles 2 and 3, respectively. The quality of the data is excellent and only some noise from the cars in Figure 5a. First breaks are clear in the merged cabled and wireless shot records suggesting that refraction data analysis and velocity tomography are possible using the data. There are also weak signs of reflections in the data, which justifies their processing and imaging as reflection seismic sections. As it will be presented later, most reflections dip towards the northeast implying

that the wireless recorders and shooting on the northern side of the road 11 was necessary to enable their imaging. Without shooting and recording there, we would not have been able to image most of the reflections observed in the data.

### 3.2 Reflection data processing

Signs of reflections in the raw shot gathers were encouraging and motivated to process the reflection component of the data. Reflections along profile 4 have for example different characters, shorter and more gently dipping, compared with those

observed along profiles 2 and 3 (Fig. 5) suggesting that the main dip favours the orientation of profiles 2 and 3 i.e. striking NW-SE and dipping NE. Table 2 summarizes the key processing steps used for the reflection data processing of the seismic data. 2D processing was employed given the straight nature of the profiles. Several factors were considered during the





processing. To avoid processing artefacts, the processing was kept simple and parameters were obtained carefully (e.g., Dehghannejad et al., 2012; Malehmir et al., 2017). The important processing steps where refraction static corrections (Figs. 5b and 6b), prestack data enhancement (Figs. 5c and 6c) and velocity analysis. Velocity analysis was important because of the steep nature of the reflections. Example raw shot gathers (from profiles 2 and 3) and how reflections were enhanced in

shot gathers are shown in Figures 5 and 6. The processed shot gathers show a number of parallel steeply-dipping reflections in both profiles. Data along the SW portion of profile 3 were excluded since they did not allow good imaging of the dipping reflections to the NE (not favouring this dip and quite noisy due to strong winds during the data acquisition). Because of this we also tested a number of ways to enhance the quality of the reflections such as the use of normal amplitude stacking and diversity-based stacking. Along profile 3 we concluded that the diversity stack provided much superior image than the

normal amplitude stack (Fig. 7). Along profiles 2 and 4, normal amplitude stack was used.

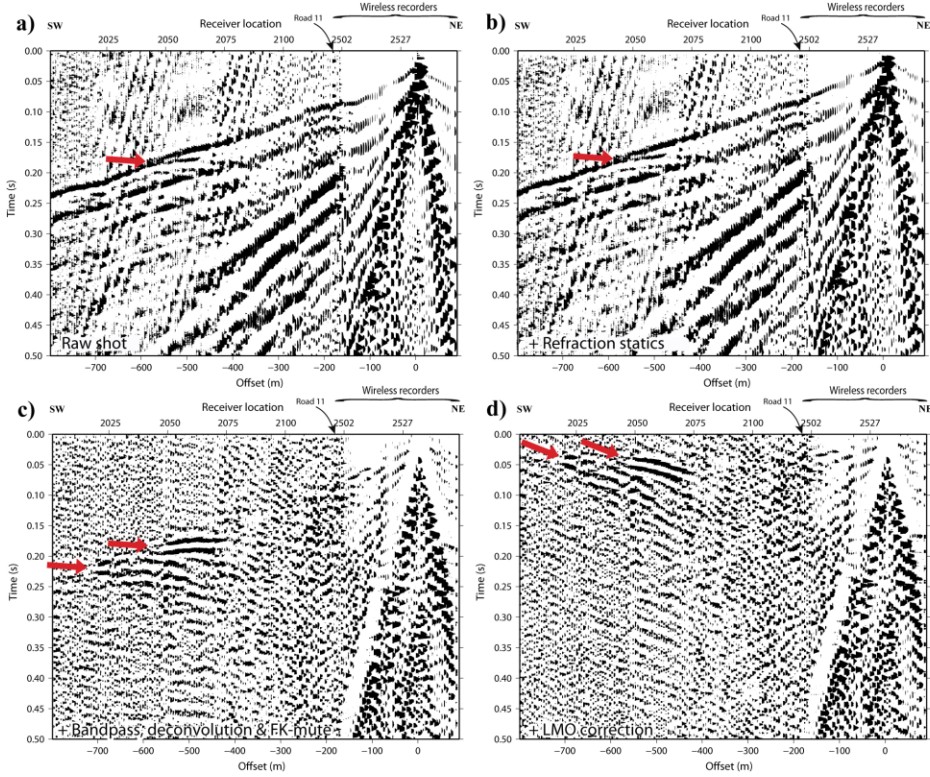

**Figure 5: An example shot gather from profile 2 (after vertical stacking of the three repeated shot records) showing the quality of the first breaks and how merged cabled and wireless data were used for both refraction data analysis and reflection data processing. Note the high level of noise from the road 11 remained as surface-wave arrivals (steep or low-velocity) but also a**

**noticeable reflection (dipping towards northeast) in the raw data but much stronger after (b) refraction static corrections, (c) prestack data enhancement (two sets of reflections) and (d) linear-moveout (LMO) corrections (for display purpose). Two sets of reflections (marked by red arrows in d) are notable between receivers 2020-2040. The application of a median filter and dip filter (FK-filter linked with linear moveout correction) helped to enhance these reflections in prestack gathers significantly.**





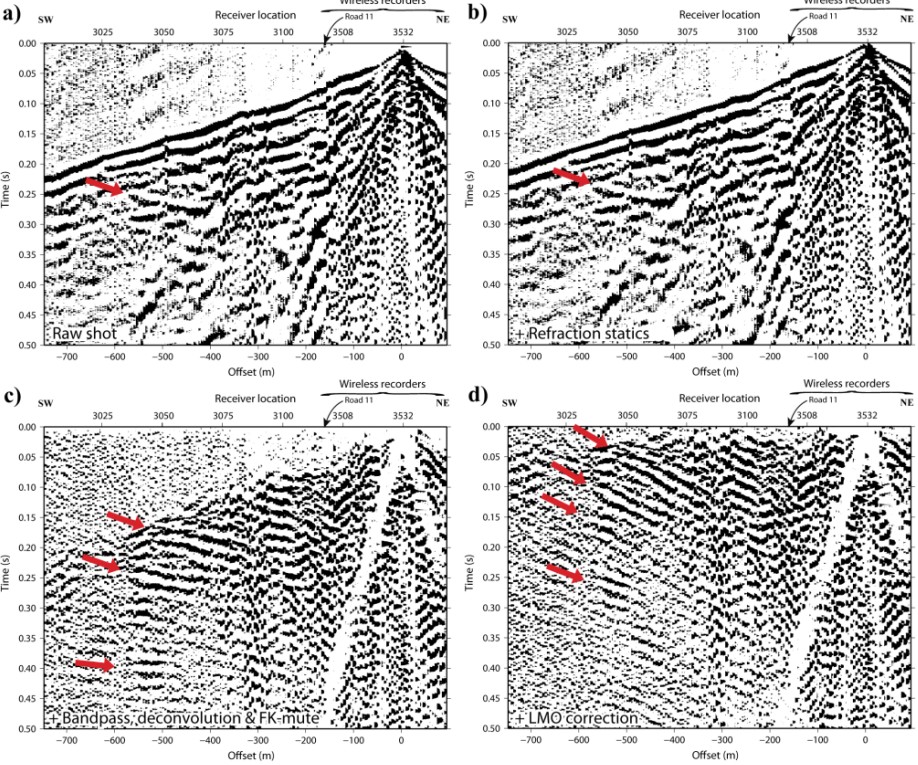

**Figure 6: (a) An example shot gather from profile 3 (after vertical stacking of the three repeated shot records) illustrating the quality of the first breaks and the data. Note a clear reflection (dipping towards northeast) already in the raw data (red arrow) but much stronger after (b) refraction static corrections, (c) prestack data enhancement, and (d) LMO corrections (for display purpose). A series of reflections (marked by the red arrows in c and d) are notable between receivers 3025-3075.**

Reflection seismic sections can be presented as unmigrated and migrated seismic sections. When reflections have considerable dip, migration should be done to bring them to their actual locations in the 2D space (assuming no out-of-the-plane effect). To migrate stacked sections velocity data are needed; we used a constant velocity of 4000 ms$^{-1}$ for both migration and time-to-depth conversion. This implies that reflections may be slightly steeper (on the order of 5 degrees) and deeper if this velocity was too low and velocities on the order of 5000 ms$^{-1}$ should have been used instead (Yilmaz, 2001). Migration was not employed for the data along profile 5 because of the gentle dip and this should be noted when the data are interpreted. Unmigrated seismic sections are shown for data quality control purposes and the reflections observed there do not represent actual locations of the geological features nor are dips true. Along profile 3, on the portion where data were excluded, prior to the migration, dummy traces were added to allow the steeply dipping reflections to be imaged in the muted region.



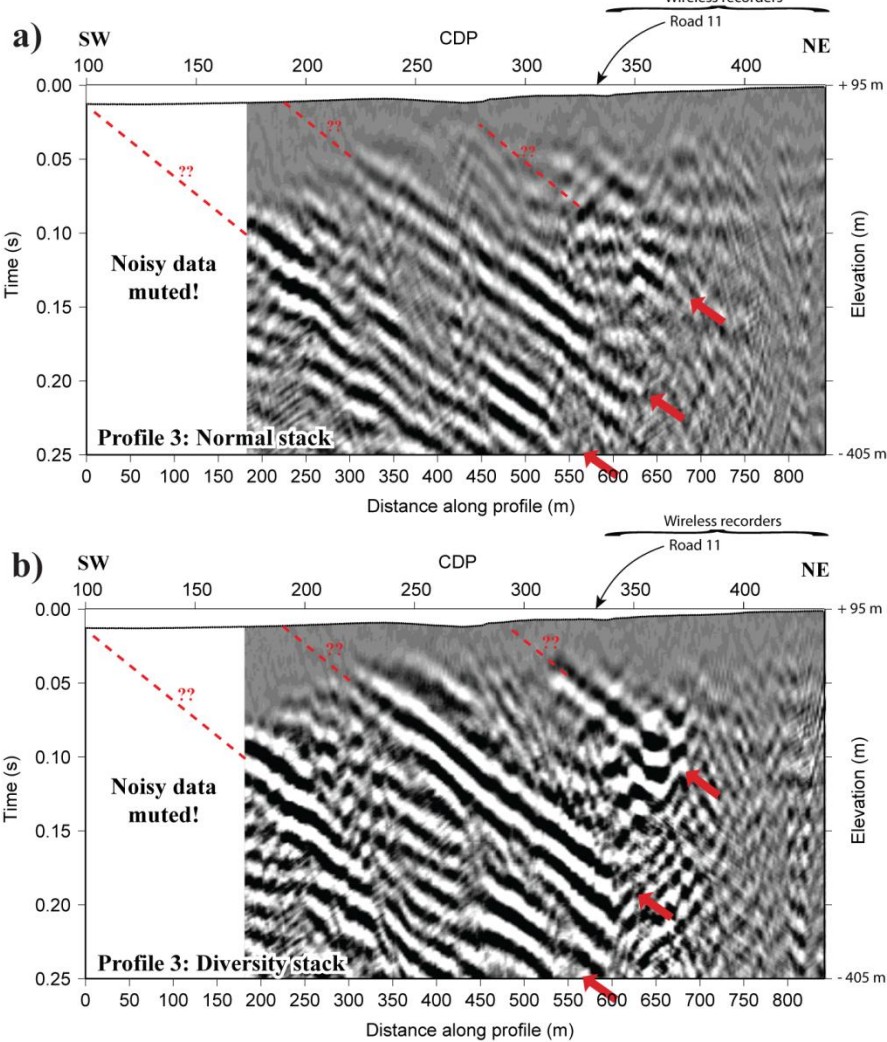

**Figure 7: Comparison between (a) normal amplitude stack and (b) diversity-based stack on the data along profile 3 illustrating much better quality imaging of the reflections obtained using diversity stack. Note that the northeastern most reflection marked by a red arrow is much notable in (b). Dashed lines provide a possible projection of the reflections towards the surface. Data on the**

5 **southernmost portion of the profile were noisy hence excluded from the stacking.**

## 3.3 Refraction data analysis and correlation with borehole data

In our study both refraction data analysis (for refraction static corrections) and tomographic methods were employed. First breaks were picked manually and corrected for where needed. We estimate an error on the order of 2 ms (based on reciprocal times) in the picking accuracy. Along profile 2 approximately 23,500 first breaks, profile 3 approximately 24,500 first

10 breaks, and profile 4 approximately 14,500 first breaks (Table 1) were used for the first break traveltime tomography (Fig. 8). A diving-wave finite-difference based 3D traveltime tomography code (Tryggvason et al., 2002 and references therein)



was used for this purpose. Cells along the profile directions were 5 m, perpendicular 10 m and at depth 2 m. Eight iterations were used to invert the data. A smooth 2D velocity model honouring the topography was used as the starting model. We used 300 ms$^{-1}$ velocity to represent the air velocity and in order to avoid rays channelling above the topography during the iterations. Smoothing constraints were used but gradually relaxed towards the final iteration numbers. The final velocity

models show RMS values around 2-3 ms as shown in Figure 8. These RMS values are acceptable given the estimated error in the picking.

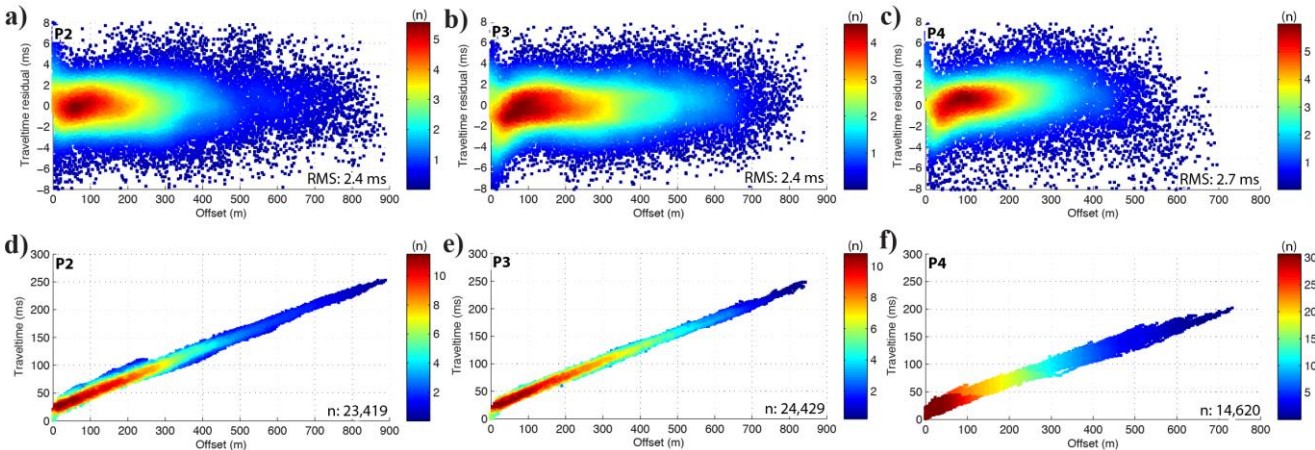

**Figure 8: Plots of the traveltime residuals (RMS misfit) obtained after 8 iterations for (a) profile 2, (b) profile 3 and (c) profile 4. Picked first breaks as a function of offsets along profiles2-4 are shown in (d-f), respectively. Note that profile 4 is shorter than**

**profiles 2 and 3.**

The final tomographic velocity models are shown in Figure 9 for all the three profiles. There is a good correspondence between low-velocity depressions and bedrock surface depressions as indicated by the drilling results conducted after the seismic survey suggesting that the tomographic velocity models can be used to constrain the bedrock surface with a good

level of confidence. Rocks in general show on average low velocities (3000-4000 ms$^{-1}$) except at a few places and in particular at a major topographic depression (ditch) in the study area (B2 and B3 in Fig. 9). This is unusual for crystalline-type rocks (Salisbury et al., 2003; Malehmir et al., 2013) and may imply an overall average-to-poor quality rocks in this area, however, it is expected from our observations in the nearby quarry and rocks situated within the highly tectonized Tornquist zone. Velocity model of profile 4 does not show significant velocity variations but shows a good correspondence at where

profile 2 is intersected but less at where profile 3 is intersected. At where profile 3 is intersected the velocity model of profile 4 better resolves the bedrock surface.

At places where bedrock is intersected at greater depths than usual, for example at 25 m depth by borehole HB5 (e.g., A2 in Fig. 9a), depression-looking bedrock is clearly observed in the tomograms suggesting a possibility for major weakness zones (likely highly fractured and/or weathered) in the bedrock.





**Figure 9: First break traveltime tomography model along (a) profile 2, (b) profile 3, and (c) profile 4 showing undulating nature of bedrock and correlation with the bedrock surfaces identified from follow-up drilling at the site. Velocity-low depressions are notable in areas labelled by A2-3, B2-3, C2-C3 and E3 particularly along profile 2. The poorest quality rocks and deepest bedrock was observed at the location of A2 (HB5) where down to 33 m depth only highly altered rocks were observed from the chips coming out of the drilling (i.e., no core drilling was performed).**

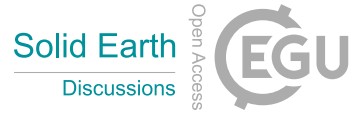

## 4 Follow-up drilling, ground magnetic survey and downhole logging

Immediately after the seismic survey and for better understanding the hydrogeological conditions (baseline) in the site, a series of boreholes (Fig. 3) were drilled. A rotary percussion-drilling rig was used for this purpose, which did not allow core sampling rather drill muds to be analysed. This choice was justified economically and for the purpose of hydrological studies requiring wide diameter boreholes. Before performing pump-tests and estimating hydrological parameters, a number of boreholes were downhole logged using natural gamma, caliper (borehole diameter), density and resistivity probes. Immediately after these the holes had to be sealed off with no possibility to conduct more surveys given their positions being in an operating farm. Downhole logging results were provided to us for this study for HB2, HB3 and HB5 (Fig. 3). For all the other boreholes depth to bedrock was provided as shown in Figure 9. Important drilling observations were also provided. We will mention this later that a suitable situation would have been to measure magnetic susceptibility and core drilling in the area. Inclined core drilling was planned to be a follow-up stage however it did not continue due to issues with land accessibility. Important for the seismic interpretations would be distinguishing Permian mafic dykes from faults and amphibolites. Caliper would be useful if severe faulting is encountered, so resistivity and density measurements. Density between dolerite and amphibolite is usually similar and they both should show low gamma radiations. Ground total-field magnetic survey using transverse type approach was also conducted and used in this study.

## 5 Results and interpretation

Several northeast-dipping, approximately 60-65 degree, reflections were imaged down to 400 m depth thanks to the close shot and receiver spacing strategy of the data acquisition. These reflections often show coherent character but at occasions are discontinuous and have different appearances. Figures 10 and 11 show unmigrated and migrated seismic sections of profiles 2 and 3, respectively. For profile 4 only unmigrated or stacked section is shown since most reflections appear at the edge of the section and nearly horizontal in the middle. Tomographic velocity models are also plotted together with the reflection seismic sections to check if velocity anomalies are associated with any of the reflections.

A careful inspection of the results suggests sub-parallel reflections dipping about 60-65 degrees towards NE along both profiles 2 and 3; they have similar characters except close to the central parts of profile 2 slightly towards its SW parts (Fig. 10b). One particular reflection is associated with the topographic depression in the study area and matches well with the high-velocity zone observed under the depression. Reflections along profile 4 have gentler dips, as one would expect, given the steep character of the reflections in the perpendicular direction (profiles 2 and 3). They may however suggest the general strike is not perpendicular to the direction of profiles 2 and 3. Reflections along profile 4 are shorter and have a disturbed character in the middle of the profile. This maybe due to the noisy nature of the data along this profile or suggest different geological structures than those observed along profiles 2 and 3. Along profile 4, some evidence of down faulting can be seen from the discontinuity of the reflections and their bivergent-wedge appearance. This may be due to the Protogin zone,





which strikes through the area in an NNE-SSW direction. Movements along this fault zone are earlier (0.9-1 Ga) than in the Tornquist zone but may very well have influenced tectonic setting in the area in conjunction with the movements alone the Tornquist zone.

Based on a series of reasons we argue the steeply-dipping reflections are mainly originated from the Permian dolerite dykes.
These include (1) the reflections are sub-parallel and favor the orientations observed in the magnetic data (Fig. 1a) and occasionally match the magnetic highs observed in the ground measurements (e.g., R2 in Figs. 10 and 11); (2) they have strong amplitudes as observed already in the shot gathers and appear in a regular order of 100 m apart (Fig. 2), this can be expected from 5-20 m thick dolerite dykes given their high density and velocity (e.g., Juhlin, 1990); (4) gneissic-amphibolitic rocks rather are part of the medium and do not appear to be isolated-individual entities to be imaged (Fig. 2a),
they are nearly equally observed in the country rock and likely can only act as a medium for other geological features; (5) with some adjustments and better matching, the reflections fit regions of decrease in natural gamma and increase in density observed in the downhole logging data (see the insets in Figs. 10 and 11). For example, along profile 2 density and natural gamma logs (Fig. 10c) show a reasonable correlation with the position of reflection R2 (5 m vertically thick zone). Resistivity logs show a drop in the resistivity but likely suggest the dyke-country-rock contact is highly water-bearing and
thus electrically conductive. Caliper log shows no variations in this region helping to exclude a possibility for a fault at this location in conjunction with a high-density zone. Borehole HB5 also shows a drop in the natural gamma (15 m vertically thick zone) associated with reflection R4 along profile 2 (Fig. 10b). Along profile 3, borehole HB3 shows even a vertically thicker region where R2 reflection may be connected to a reduced natural gamma zone (Fig. 11c), likely a dolerite dyke.





**Figure 10: (a) Ground total-field magnetic, (b) unmigrated and (b) migrated (also time-to-depth converted) reflection seismic sections along profile 2. Tomographic velocity model is projected onto the migrated section for direct comparison. Red arrows mark the main reflections. They may extend at depth but we are unable to image them due to the short length of the profile. Downhole logging results from boreholes HB2 and HB5 suggest that the reflections R2 and R3 are associated with zone of reduced natural gamma, high density and low resistivity. Caliper log suggests no change in the borehole diameter. Based on these we argue that the reflections are originated from dolerite dykes.**





**Figure 11: (a)** Ground total-field magnetic, **(b)** unmigrated and **(b)** migrated (also time-to-depth converted) reflection seismic sections along profile 3. Tomographic velocity model is projected onto the migrated section for direct comparison. Red arrows mark the reflections. They may extend at depth but we are unable to image them due to the short length of the profile. Downhole logging results from borehole HB3 suggest that the reflection R2 is associated with zone of reduced natural gamma. Caliper log suggests no change in the borehole diameter. Based on these we argue that the reflection is originated from a dolerite dyke.





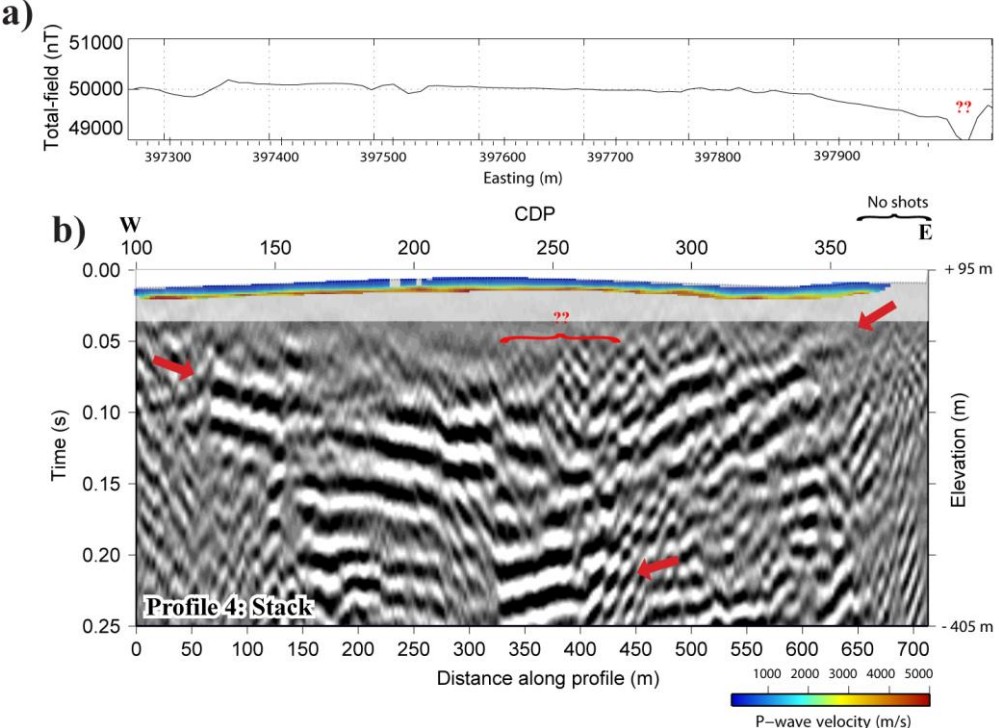

**Figure 12: (a) Ground total-field magnetic, and (b) unmigrated (but time-to-depth converted) reflection seismic section of profile 4. Tomographic velocity model is projected onto the migrated section for direct comparison. Red arrows mark the reflections that appear to be divergent and in the middle of the profile more horizontal. The central part of the profile shows indications of down faulting.**

To help better understand 3D geometry of the reflections, we first visualized them in 3D and then picked the reflections as observed in the migrated seismic sections (Fig. 13a). Visualization was done in conjunction with the tomography results (Fig. 13b). Again, a careful inspection of this figure suggests a major NW-SE trend for most of the reflections and this strike is consistent with the dyke swarms formed during the Permo-Carboniferous break-up of Pangaea in Scania (Obst and Katzug, 2006) as shown in Figure 1. Based on these we identified four major sets of reflections (R1-R4; Fig. 13) that we interpret to be from dolerite dykes.





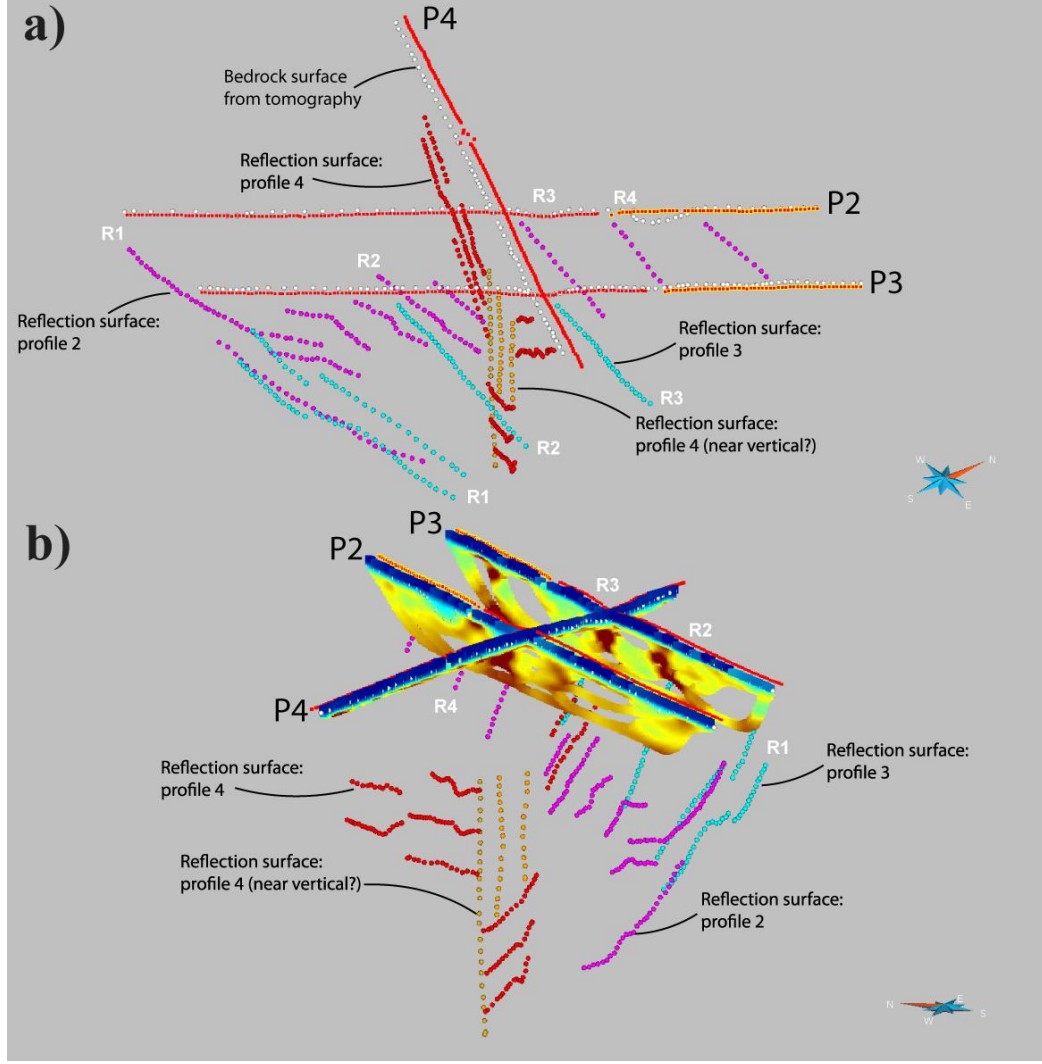

**Figure 13: 3D views showing the interpreted bedrock surface (white points), reflections from profile 2 (purple points), profile 3 (blue points), and profile 4 (red and brown points (likely vertical faults)). Note that the red points on the surface represent the actual locations of the seismic sensors.**

5  **6 Discussion**

A fundamental surprise in this study is why the dolerite dykes, if our interpretation is correct, are dipping and not sub-vertical as observed in most quarries north in the study area (Fig. 2). Through a series of arguments we further discuss this and what this may imply in terms of geology and its significance for developing the thermal storage site in the study area.



## 6.1 Hydrogeology of the site

Two different hydrological measurements were conducted to check underground water flow in the study area. The first one was measuring the groundwater table in the sediments through existing boreholes and wells (Fig. 14a), and the second one was measuring groundwater table in the bedrock through boreholes that intersected the bedrock (Fig. 14b). This was

motivated to check if any particular pattern in the groundwater flow can be associated with the geological structures in the area. As one can expect, the measurements in the sediments (Fig. 14a) suggest a ground water flow towards lower elevations in the area with the central part of profile 4 being on a hill (red contours). The measurements in the bedrock however provided much more appealing results and a pattern for the groundwater flow towards SE (Fig. 14b) and an orientation consistent with the directions of the dykes and major structures in the area (Fig. 1a). Reflections (R1-R4) greatly match this

orientation although this study cannot alone prove that the dykes can be steeply dipping or sub-vertical; both scenarios would satisfy such an observation.

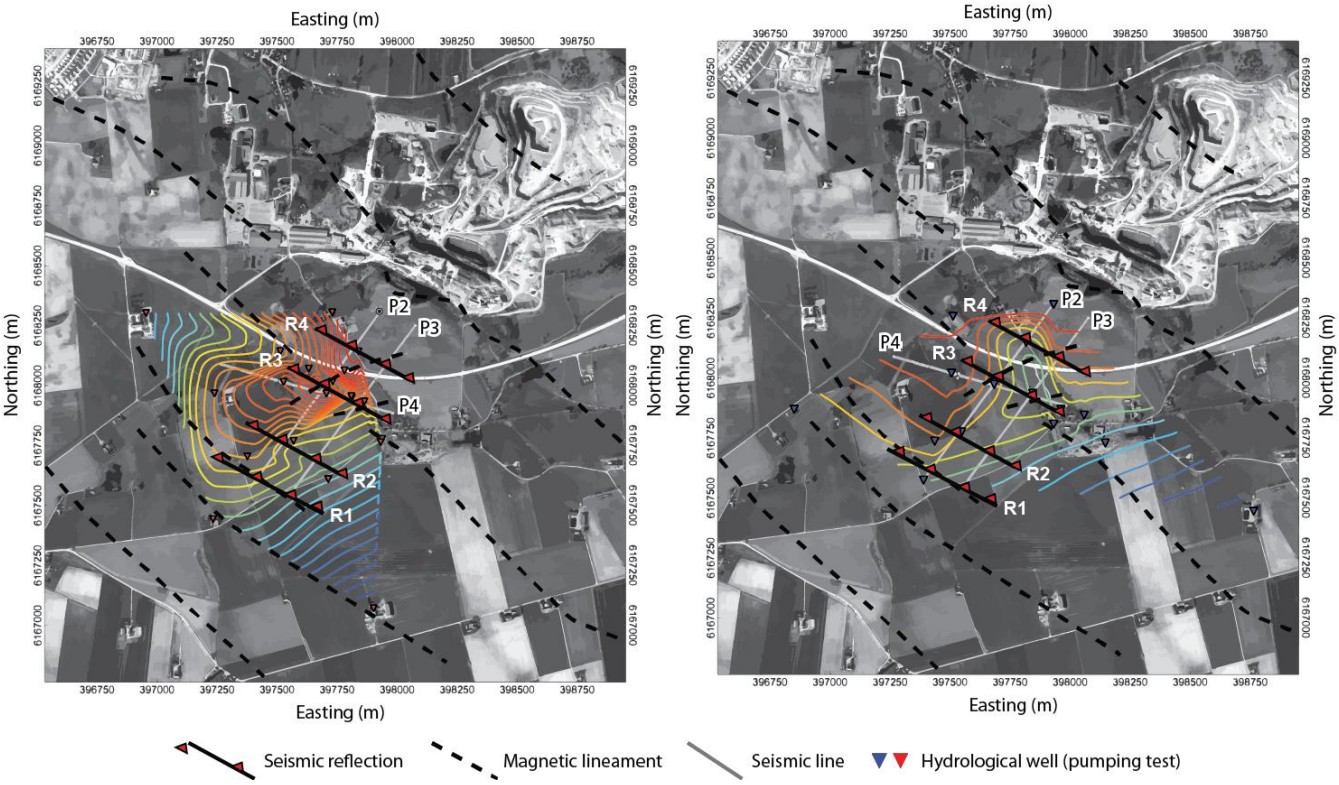

**Figure 14: Hydrological measurements conducted at the site to better understand groundwater flow in (a) sediments and (b) bedrock and if they can be connected to any underlying geology controlling their patterns. Red contours indicate high water table**

**and blue lower water table. In the case of bedrock, a clear NW-SE groundwater flow is notable, which may indicate the role of the steeply-dipping dykes in this. Labelled as R1-R4 are the reflections observed in the seismic data. Dashed lines are the peaks of the magnetic lineaments extracted from the aeromagnetic data from the study area.**



## 6.2 Magnetic susceptibility transects across the dykes

Because no further downhole logging was possible in the boreholes drilled in the study area, we decided to revisit the rock quarry north of the site (Figs. 2a and 3) to conduct a series of magnetic susceptibility measurements in order to check if amphibolites and dolerite dykes have comparable magnetic properties. Dolerite dykes showed much stronger magnetic

susceptibility (Fig. 15) than the amphibolites, hence should be identifiable on this basis, their magnetic properties (as also evident on the magnetic map). One important observation is also that the amphibolites tend to generally gently dip towards SW (Fig. 15) and this is inconsistent with the reflection orientations we observed in the data. The two dolerite dykes observed in Figure 15 crosscut the gneissic and amphibolitic rocks and the major one (approximately 5 m thick) appears also be dipping at about 60-65 degree towards the NE. In these measurements two important observations were made namely (1)

the Permian dykes have occasionally dip character similar to that of the seismic data i.e. not necessarily sub-vertical and (2) they are the only magnetic features in the area.

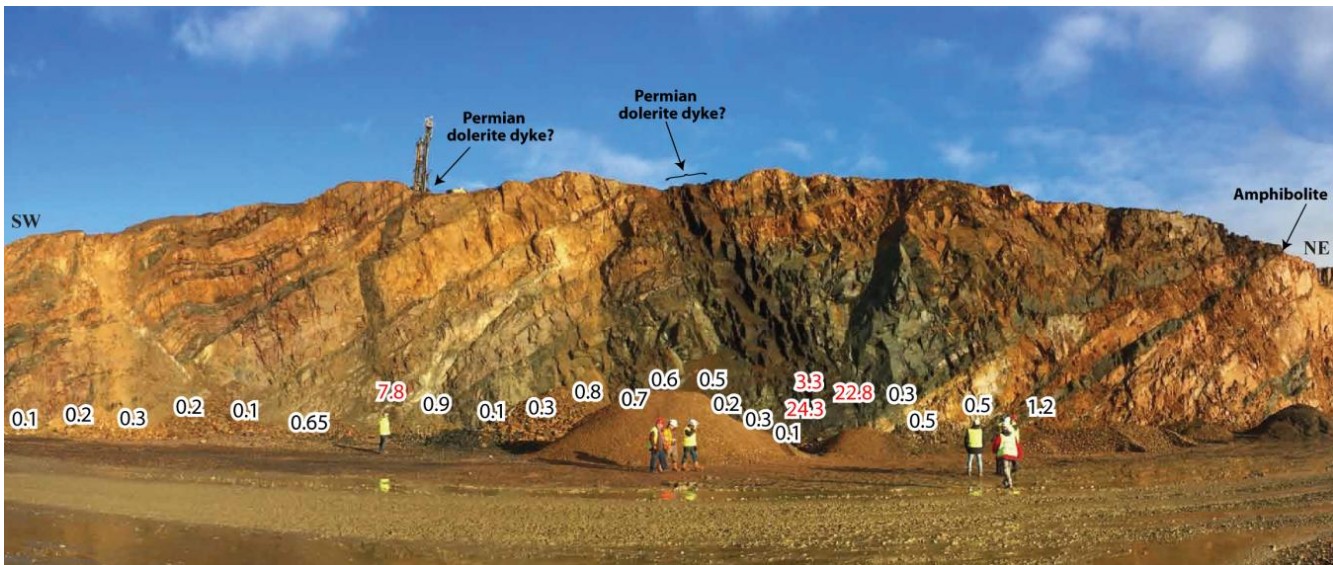

**Figure 15: Magnetic susceptibility measurements (average of 3-5 readings at each point is shown) across a section of the rock quarry north of the study area showing how strongly magnetic (3-30 order higher) the Permian dolerite dykes are compared to the**

**other country rocks (e.g., amphibolite and gneiss) in the area. The main dyke observed here has a NE dip component similar to those observed in the reflection seismic data. Magnetic susceptibilities are shown $10^{-3}$ SI unit (e.g., 22.8 becomes 0.0228 in SI unit). An SM20 handheld susceptibility meter was used for this purpose.**





## 6.3 Revisiting BABEL offshore seismic lines A-AA-AB

It became immediately evident to us to check if other similar types of datasets can also provide information on the deeper geometry of the Permian dykes within the Tornquist zone. Apart from the work presented by Phillips et al. (2017) from the

southern Norwegian North Sea, we decided to revisit the BABEL offshore seismic data from the Hanö Bay Basin and check if any evidence of dyke emplacement could be found in the Precambrian basement. Sediments in the Hanö Bay Basin are of Mesozoic age and therefore cannot contain any Permian dykes. They can, however, provide information on faulting history within the Tornquist zone. Processed stacked sections from 1990 (BABEL Working Group, 1990) from lines A-AA-AB were merged and used for this purpose. Low-frequencies remained from the original work were first filtered out, coherency of the reflections improved using a FX-deconvolution filter, and then migration using a 2D velocity model attempted. To

quality assure the migration process we used a number of diffractions observed on the southwestern parts of the merged section as well as the rough geometry of the basement producing edge diffractions or bow-ties and examined if they collapsed to a point and did not look too smiley. This was in particular important for the area under the Hanö Bay given the rapid change from sedimentary to crystalline rocks. Figure 16 shows the resulting image for the whole section of 13 second data, and the top 2 second for the Hanö Bay Basin region. While multiples are still evident in the image, the Precambrian

basement manifests itself as a highly undulating surface (Fig. 16a). Two short segments of reflections are clearly notable around 700 ms time (between the red dashed lines) and likely have of basement origins (about 200 ms or approximately 200-300 m throw). The faults also appear to show a multi-phase history with both compression (thrusting) and extension (normal faulting). Apart from this, there is no evidence of steep dykes in the basement.

Looking into the deeper parts of the section (Fig. 16b), we clearly realize the Moho definition (transition from a highly reflective lower crust to transparent solidified, seismically homogenous upper mantle) around 12 second (approximately 40 km depth) and a clear Moho step (keel of approximately 5-10 km high and 30-40 km wide) under the Hanö Bay Basin (slightly inclined towards its southwest half). While there are a number of SW-dipping reflections in the lower-middle crust projecting towards the location of the Sorgenfrei Tornquist Zone, we cannot be sure if these are dykes or frozen magma

chambers where dykes intruded to the upper crust within the Tornquist zone. Some strong dipping reflections are also evident under the Hanö Bay Basin in the crystalline middle-upper crust but again not evident if they can be connected to the dolerite dykes within the Tornquist zone in similar character and geometry as observed on the magnetic data or they represent local dykes intruded into the middle-upper crust.





**Figure 16: Reprocessed (only poststack processing) and migrated seismic section of BABEL offshore lines A-AA-AB (BABEL Working Group across the Sorgenfrei Tornquist Zone (a) Hanö Bay Basin area. The whole seismic section is shown in (b) including a clear Moho step under the Hanö Bay Basin. Clear faults are evident in the Hanö Bay Basin and show a throw as much as 200-500 m and a multi-phase history (both reverse and normal faulting). The original work did not present such a clear Moho keel and migration was done on line drawings likely using a 1D or constant velocity model (BABEL Working Group, 1990).**

## 6.4 Tectonic model and implication for thermal storage

There are two scenarios that may explain imaging the Permian dykes in the seismic data. Firstly, they are not sub-vertical everywhere within the Tornquist zone and they quickly turn steeply dipping in the subsurface as also observed at a few



quarries in the area (e.g., Fig. 15; see also Sivhed et al., 1999). Secondly and our preferred scenario, the region immediately south the quarry is strongly influenced by the Romeleåsen Fault Zone (thrusting during Cretaseous-Neogene compressive tectonic regime) so that it has led to a block rotation of the basement rocks including the Permian dykes (Fig. 17). This has turned the dykes from being sub-vertical in most places to steeply-dipping and tilted towards NE. The study area is located

nearly at the edge of the Romeleåsen Fault Zone and hence this is likely possible that the seismic profile crosses the fault zone or is very close to it. Such a scenario is also consistent with what is suggested by Sivhed et al. (1999) and Erlström and Sivhed (2001) that immediately south of the Romeleåsen Fault Zone, faults and dykes dip towards NE as a result of a basement tilting. Again for the reasons mentioned earlier we think that the reflections are originated from the Permian dykes and not faults nor amphibolite lenses. Figure 17 provides a tectonic sketch of our preferred scenario illustrating this

interpretation.

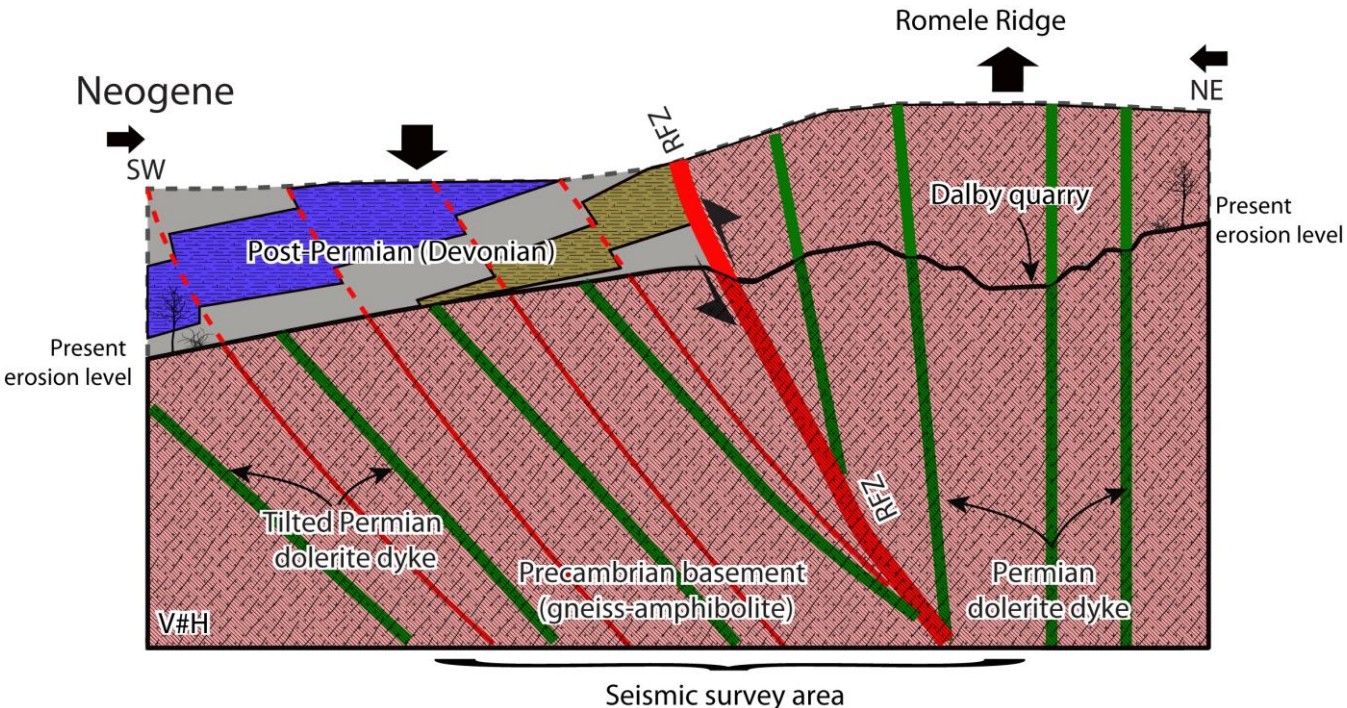

**Figure 17: A simplified tectonic model (Neogene-now) illustrating our preferred scenario (see also Erlström et al., 2001) showing why the Permian dykes exhibit steeply-dipping character south the Dalby rock quarry. We associate this with a block rotation of**

15 **the basement rocks immediately south of the quarry or near to it (situated in the Romele Ridge) that led to the tilting of the dykes from sub-vertical to steeply-dipping towards the northeast. Note that we expect some sedimentary rocks to be present south of the Romeleåsen Fault Zone.**

A main implication of our interpretation of the dipping dykes instead of sub-vertical is that the storage site (if greater than 100-200 m diameter) may be intersected by one of these dykes. Given that the dyke-country rock contacts are likely highly





water bearing, they can have great influence on the underground water flow and hydrogeology of the subsurface. If these dykes intersect the storage, the dykes would not then act as a barrier for the water (to contain the water or fluids to be used for the thermal energy storage) rather complicates potential leakage and fluid removal in case any leakage occurs. If building a storage site would become a reality again, core drilling, a 3D seismic survey and appropriate downhole logging (including

magnetic susceptibility) should be done. Crosshole and VSP measurements can also be done to provide greater details in the subsurface. Given that Phillips et al. (2017) and this study both observe the dolerite dyke swarm N-NE dipping, large-scale high-fold seismic transects across the Tornquist zone would be important since they may imply a major magma source region towards the south feeding a more than 500-km radius dyke swarm system (Fig. 1).

## 7 Conclusions

In this study we have acquired three high-resolution seismic profiles with a primary objective of characterizing an underground thermal energy storage site in southwest of Sweden within the Sorgenfrei Tornquist Zone. A number of boreholes were drilled following the seismic study and provided a better ground for the interpretation of the seismic data. Clear refracted and reflected arrivals were identified in the data and allowed a clear delineation of bedrock surface and structures within it that are important for the geological storage and tectonic evolution of the site. We argue that majority of

the reflections, 60-65 degrees NE-dipping, are originated from Permian dolerite dykes some of which acted as planes where thrusting occurred. They appear in regular order and consistent from one profile to another. Given that in most places these dykes are observed sub-vertical, these seismic reflection images would then be first time showing them in such a steeply-dipping character. We think the steeply-dipping dykes have been tilted from sub-vertical due to the thrusting of the Romeleåsen Fault Zone immediately located south of the Dalby rock quarry. Its implication for locating a thermal storage

can be that these dykes would then be encountered if the storage is larger than 100 m radius. The dyke-country rock contacts are likely highly water bearing and hydraulically conductive, hence may not act as a barrier rather a leaking zone. Most of these dykes have also intruded into the faults formed during extension/rifting phases within the Tornquist zone implying to be potentially zones of weaknesses for construction purposes.

To complement this study, historical offshore BABEL seismic data across the Sorgenfrei Tornquist Zone were also revisited.

The new processing work conducted on poststack sections of lines A-AA-AB led to better recognition of the Moho and its necking character (5-10 km of Moho step across 30-40 km wide zone) under the Tornquist zone. While several others have suggested this, it is greatly evident in the revisited seismic section in this study because of better handling poststack migration and improving coherency of the reflections. A number of faults and their kinematics (normal and reverse) have also been inferred within the Hanö Bay Basin providing compelling evidence of multi-phase history of the Tornquist zone

(tectonic inversion).





**Acknowledgements**

This work was supported by Skanska, and benefited collaborations among experts from Sweco, Lund University and Skanska. Trust project (http://www.trust-geoinfra.se) supported mainly by Formas (2012-1907), BeFo, SBUF, Geological Survey of Sweden (SGU) and Skanska helped to initiate this project. We thank Skanska for collaborating with us in this project, also PhD and MSc students from Uppsala University who took parts in the data acquisition and preparation of the data. This project was initiated as a spin-off from Trust-GeoInfra (http://trust-geoinfra.se/) project where both Skanska and Uppsala University were partners in Trust 2.2 subproject. BABEL data along lines A-AA-AB were revisited through a Swedish Research Council (VR) funding (2015-05177) for which we are thankful. We thank R. Hobbs for providing access to the original BABEL seismic lines.

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

**Table 1. Main seismic acquisition and refraction parameters of the Dalby data, August 2015.**

| Spread parameters | Profile 2 | Profile 3 | Profile 4 |
|---|---|---|---|
| Recording system | Sercel Lite | Sercel Lite | Sercel Lite |
| Survey geometry | Fixed | Fixed | Fixed |
| No. of receivers | 172 (2001-2122, 2501-2551) | 166 (3001-3113, 3501-3551) | 141 (4001-4141, 4130-4141) |
| No. of shots | 170 (3 records/point) | 157 (3 records/point) | 120 (3 records/point) |
| Shot/receiver spacing | 5 m | 5 m | 5 m |
| Maximum offset | ~ 900 m | ~ 850 m | ~ 800 m |
| Source type | 500-kg Bobcat drophammer | 500-kg Bobcat drophammer | 500-kg Bobcat drophammer |
| Geophone | 10 Hz, spike | 10 Hz, spike | 10 Hz, spike |
| Sampling interval | 0.5 ms | 0.5 ms | 0.5 ms |
| Record length | 20 s (1 s used for processing) | 20 s (1 s used for processing) | 20 s (1 s used for processing) |
| Wireless data harvesting | GPS time | GPS time | GPS time |
| Total no. of traces | 29,240 | 26,062 | 16,920 |
| Geodetic surveying | DGPS | DGPS | DGPS |
| **Refraction parameters** | | | |
| No. of first breaks | 23,419 | 24,429 | 14,620 |
| Method | 2D ray-tracing and 3D first break traveltime tomography | 2D ray-tracing and 3D first break traveltime tomography | 2D ray-tracing and 3D first break traveltime tomography |
| Cell sizes (x,y,z) | $5 \times 10 \times 2$ m | $5 \times 10 \times 2$ m | $5 \times 20 \times 2$ m |
| No. of iterations | 8 | 8 | 8 |
| RMS | 2.4 ms | 2.4 ms | 2.7 ms |





*Table 2. Key reflection processing steps (Globe Claritas$^{TM}$ processing software was used for the processing).*

| Steps | Profile 2 | Profile 3 | Profile 4 |
|---|---|---|---|
| 1 | Read SEGD data | Read SEGD data | Read SEGD data |
| 2 | Zero-time correction (automatic and manually corrected) | Zero-time correction (automatic and manually corrected) | Zero-time correction (automatic and manually corrected) |
| 3 | Vertical shot stacking | Vertical shot stacking | Vertical shot stacking |
| 4 | Geometry setup (CDP spacing 2.5 m) | Geometry setup (CDP spacing 2.5 m) | Geometry setup (CDP spacing 2.5 m) |
| 5 | First break picking | First break picking | First break picking |
| 6 | Refraction static corrections | Refraction static corrections | Refraction static corrections |
| 7 | Elevation static (4000 ms$^{-1}$, 95 m) | Elevation static (4000 ms$^{-1}$, 95 m) | Elevation static (4000 ms$^{-1}$, 95 m) |
| 8 | Band pass filter (20-40-180-220 Hz) | Band pass filter (20-40-180-220 Hz) | Band pass filter (20-40-160-280 Hz) |
| 9 | Deconvolution (22 m gap) | Deconvolution (22 m gap) | Deconvolution (22 m gap) |
| 10 | FK filter | FK filter | FK filter |
| 11 | Median filter (linked to NMO) | Median filter (linked to NMO) | Median filter (linked to NMO) |
| 12 | AGC (200 ms) | AGC (200 ms) | AGC (200 ms) |
| 13 | Residual static corrections | Residual static corrections | Residual static corrections |
| 14 | NMO corrections (30% stretch mute) | NMO corrections (30% stretch mute) | NMO corrections (30% stretch mute) |
| 15 | Stack (normal) | Stack (diversity) | Stack (normal) |
| 16 | FX-decon | FX-decon | FX-decon |
| 17 | Balance amplitude | Balance amplitude | Balance amplitude |
| 18 | Padding | Padding | |
| 19 | Migration (finite difference, 4000 ms$^{-1}$) | Migration (finite difference, 4000 ms$^{-1}$) | |
| 20 | Time-to-depth conversion (constant 4000 ms$^{-1}$) | Time-to-depth conversion (constant 4000 ms$^{-1}$) | Time-to-depth conversion (constant 4000 ms$^{-1}$) |
| 21 | Export for plotting and 3D visualization | Export for plotting and 3D visualization | Export for plotting and 3D visualization |

