# Peer review of "Seismic imaging of dyke swarms within the Sorgenfrei Tornquist Zone (Sweden) and implications for thermal energy storage"

_Solid Earth, 2018_

## Short Comment (SC1) · 25 Sep 2018

I found this a very interesting read, partly because the combination of techniques and disciplines is novel to me. I have three comments, one slightly pedantic and two key observations that I think has been overlooked.

1) On page 4, Line 27 you state:

"Although a matter of a debate, Phillips et al. (2017) claim the first dyke swarm images observed in reflection seismic data in the world."

In Phillips et al (2017) [for transparency I should clarify I'm an author on this paper], it

is stated:

"While previous seismic-based studies have imaged or contain evidence of one or several dikes, which may or may not be part of a dike swarm (e.g., Zaleski et al., 1997; Malehmir and Bellefleur, 2010; Wall et al., 2010), we present the first seismic data set that images and constrains the geometry of a dike swarm."

To clarify our sentence construction here, we agree that other seismic-based studies have clearly identified individual dykes with a dyke swarm (e.g. Wall et al. 2010 image three or four dykes). The orientation of these dykes does provide information on the orientation of the dyke swarm, but it does not constrain other geometrical properties (e.g. width, changes in height) of the dyke swarm. As such, and after an extensive literature search, we think the data presented in Phillips et al. (2017) are the only data that images the entire width of a dyke swarm and does so for >100 km along-strike; thus providing a novel insight into the 3D geometry of a dyke swarm. However, please clarify why you consider this a matter of debate. If we have missed pieces of literature that seismically image the structure of dyke swarms (not just individual intrusions) then we would be very grateful if you could cite them to bring this work to light and make up for any oversight in Phillips et al. (2017).

2) A key observation when considering whether dykes intersect storage sites will be spacing between dykes. You clearly highlight that dip and thickness are important. From your images I think you could also say something about spacing. Bunger et al. (2013) [Bunger, A.P., Menand, T., Cruden, A., Zhang, X. and Halls, H., 2013. Analytical predictions for a natural spacing within dyke swarms. Earth and Planetary Science Letters, 375, pp.270-27] show that dyke spacing should be systematic. The size of the storage site relative to the spacing between dykes could thus be a determining factor in its placement. From all your different data sources, it should be possible to put some quantitative constraints on dyke spacing.

3) There seems to be no mention of what the resolution of the data is. You mention

Interactive
comment

it is 'high-resolution' but there should be some consideration of what this means; in other words, what are you missing? E.g. are <5 m thick dykes imaged and, if not, what could their importance be? There should also be an explanation about tuned reflection packages.

Kind regards, Craig Magee

———————————————

---

## Referee Comment (RC1) · Anonymous Referee #1 · 3 Oct 2018

The paper presents results from three high-resolution 2D seismic lines acquired as a first stage characterization of the subsurface for a potential thermal energy storage site in Sweden. The focus of the paper is really on dolerite dykes which are important geological features in the study area. The topic is of interest and relevant to this special issue. In my opinion, the paper should be published after some weaknesses and loose ends are properly taken care of during revision (see main and detailed comments below).

Main comments:

1) The paper either was written quickly or did not deserve all required attention prior

to submission. The language is often imprecise and sometimes sentences are not grammatically correct. This is not systematic through out the manuscript but prevalent enough to require proper and careful editing by a native English speaker.

2) The correspondence between log anomalies and reflections (R2 and R3) is particularly good on the un-migrated section (Figure 10b) but rather poor on the migrated section (Figure 10c, especially for R3). What makes you think that the anomalies on logs are really the signature of the dolerite dykes if logs and reflections are not at the same spatial location? Are there any other possible explanations for these log anomalies? The same observation applies to figure 11 (i.e. poor fit with log anomalies and reflection R2 on migrated section). How confident are you about the migration velocities and velocities used for T-Z conversion?

3) I suggest looking for references on physical rock properties of dolerite dykes to support the interpreted signature on logging data (especially low natural gamma) and seismic data. Specifically, the acoustic impedance of dolerite dykes and potential contrast with host rocks are not discussed in the paper. However, acoustic impedance is the only property that can unequivocally confirm the reflective (i.e., P-wave) nature of dolerite dykes. The interpretation of steeply-dipping reflections as dolerite dykes currently lacks that irrefutable argument (even though strong conjectural arguments are provided to support this interpretation).

4) The paper includes a discussion on the implications of results for thermal energy storage without really providing the key characteristics of a good site for such storage. Some details about an optimal thermal storage site should be provided. For instance, what is the size of such a site ("caverns")? Why is this area considered suitable for thermal storage (particular rock types, geomechanical properties, close proximity to city)? It is difficult for readers to assess the implications of results for something that is not properly defined.

5) Results from the Babel seismic lines are of certainly of interest but primarily from

a background perspective. It is not clear how results from these lines effectively help the interpretation of results at the local scale or provide information useful for the discussion. Because of this, and unless a proper case can be made, I suggest to include the Babel lines as background information as a component of the geology (i.e., deep geology in this case).

Detailed and technical comments:

Page 1 line 19: dykes cannot "express themselves"... what about "are"

Page 1 line 19. "...express themselves mostly sub-vertical. They can therefore act as a good water/fluid barrier" It is not clear why the fact that dykes are vertical makes them good water barrier. Obviously, more information is needed to support this statement. Geometry is certainly an important factor but not the only one. Please provide evidences supporting that dolerite dykes are effective water barrier.

Page 2 line26: "...and if major bedrock undulations" with "...and determine if major bedrock undulations"

Page 6 line 1: "where geological structures are favourable" please be more specific about the geological structures.

Page 8 line 2: replace "where" with "were"

Page 8 line 9: What explains the better results with diversity stack over conventional stack on line 3?

Page 9 line 12: "Migration was not employed for the data along profile 5". Do you mean profile 4?

Figure 12 shows mostly dipping reflections, not steep but still dipping reflections. The migrated section should also be shown in the paper for a "proper" positioning of reflectors.

Page 10 line 8: replace "for where needed" with "where needed"

Page 11 line 2: "A smooth 2D velocity model". Please provide more information about this model (lower and upper velocities and their distribution (ie. constant gradient?)).

Page 11 line 14: "suggesting that the tomographic velocity models can be used to constrain the bedrock surface with a good level of confidence". What velocity is the threshold to determine the bedrock surface? In addition, I suggest drawing the top of bedrock on Figure 9 (with dashed line or other).

Page 11 line 23: "...depression-looking bedrock is clearly observed in the tomograms suggesting a possibility for major weakness zones (likely highly fractured and/or weathered) in the bedrock." There are other supporting evidences for this elsewhere in the manuscript. I suggest including them here (for example, caption of figure 9 mentions cuttings from bedrock with alteration). What about velocities from tomography – can they help? Also, could you please be more specific about the alteration observed in cuttings?

Page 13 line 15: "using transverse type approach". What do you mean specifically? Survey lines orthogonal to the main structures?

Page 13 line 17: "Several northeast-dipping, approximately 60-65 degree, reflections were imaged down to 400 m depth thanks to the close shot and receiver spacing strategy of the data acquisition." Not sure I understand what is meant by "thanks to close shot and receiver spacing". Aren't large offset required to image steeply dipping reflectors?

Page 13 line 19: "but at occasions are discontinuous and have different appearances". Please be specific about the different appearances.

Page 13 line 25 "One particular reflection is associated with the topographic depression in the study area and matches well with the high-velocity zone observed under the depression." This is somewhat contrary to what was said before on bedrock depressions corresponding to faults and/or alteration which would normally have low velocity.

Please clarify this statement or earlier statement about depression.

Page 13 line 29: "suggest different geological structures than those observed along profiles 2 and 3". What would be the different structures and why can't they be observed on the two other profiles? Explanations that follow in the 4 lines after are rather vague.

Page 14 line 10: "..they are nearly equally observed in the country rock and likely can only act as a medium for other geological features.." Not clear at all what you mean. . .

Page 14 line 11: "with some adjustments and better matching. . .". What adjustments and better matching? Please clarify? See also main comment #2.

Page 14 line 18: ". . .connected to a reduced natural gamma zone (Fig. 11c), likely a dolerite dyke." Why would dolerite dykes have low natural gamma? Any reference to support this?

Page 19 line 8: is "appealing" the right word?

Page 19 line 8: "a pattern for the groundwater flow towards SE (Fig. 14b) and an orientation consistent with the directions of the dykes and major structures in the area (Fig. 1a). Reflections (R1-R4) greatly match this orientation". I would argue that contour lines presented on figure 14b are more complex than described in the statement above which appears a bit oversimplified. Please provide a more precise description and location (i.e. between R3 and R4 near P2 and P3?). Also, why are lines of water table continuous across reflection R1, R2, R3, and R4 if these reflections are water barriers?

Page 21 line 21: ". . . transparent solidified, seismically homogenous upper mantle" Could you please explain the meaning of solidified in terms of seismic characteristics of the crust?

Page 21 line 23: "While there are a number of SW-dipping reflections in the lower-middle crust projecting towards the location of the Sorgenfrei Tornquist Zone, we cannot be sure if these are dykes or frozen magma chambers where dykes intruded to the upper crust within the Tornquist zone." Agree. In fact, there are many possible explanations for those reflections. It might be worthwhile to add a few rather than just providing dyke-related possibilities (i.e., dykes and magma chamber that fed dykes).

Page 22 line 9: "Firstly, they are not sub-vertical everywhere within the Tornquist zone and they quickly turn steeply dipping in the subsurface as also observed at a few quarries in the area (e.g., Fig. 15...". Could you please clearly indicate what you are referring to in Figure 15 (may be use double arrows – one at the top of cliff and the other at the base to point to the feature you are referring to on Fig. 15)?

Figure 1. Please add coordinates to a) and b).

Figure 2. It would be useful to add viewing direction for a) and b) or general orientation of the pictures.

Figure 13: The 3D perspective view is very difficult to visualize on this figure (especially for a). I suggest adding axis (x,y,z) that would improve the perspective view.

Figure 14 please add a) and b).

---

## Referee Comment (RC2) · Anonymous Referee #2 · 11 Oct 2018

The paper nicely illustrates the ability of the seismic method to contribute to the geological evaluation of the thermal energy storage site. The acquisition and processing methodic, parameters and the workflow described clearly in the text, although the actual delineation of the bedrock surface from the tomography is vague. Also, section 6.3 concerning the entire crust structure seems a bit detached from the main part and objectives of the paper. I would recommend the work for publication with some corrections.

Here are my comments and suggestions:

Page 2: "The seismic survey had an initial objective of identifying depth to bedrock

and if major bedrock undulations could be related to zones of weaknesses (fractured or/and altered) in bedrock." I did not see the comparison of the depth to the bedrock from seismic (diving wave tomography) to drilling results in the paper. You provided some of it in Fig. 9 but did not summarize the results and did not provide any errors bars or discrepancies. Can you clarify this part please?

Page 7, line 10: "that the wireless recorders and shooting on the northern side of the road 11 was necessary to enable their imaging" Shooting – yes, but recording did not provide much looking at the shot records in Figs 5 and 6. All the north-dipping reflections in the records are present in the cable part. Wireless part did not contribute much due to short offsets, unfavorable north dip and maybe due to greater noise as indicated in the stacked sections in Figs 7 and 11. To justify this statement you can show some shot gathers from the source locations in the southern ends of profiles 2 and 3. The wireless part did record very good first arrivals, though.

Page 9: "we used a constant velocity of 4000 ms-1 for both migration and time-to-depth conversion" and later "5000 ms-1 should have been used instead" Why 4000m/s velocity was used? According to Fig 9, tomography velocities are closer to 5000m/s at the bedrock (and deeper down) and might be even greater for amphibolite, gneiss and dolerite.

Page 9, line 12: meant profile 4, not profile 5?

Figure 8: Please indicate what do the colors mean?

Page 14: These include (1)... (2) they have... (4) gneissic-amphibolitic (3) is missing

Figure 13: The relations of the features in the 3D view are unclear and impossible to interpret – looks like a 2D image. Perhaps it would be better to plot X-Y-Z axes and connect reflection points by wired surfaces.

Section "6.3 Revisiting BABEL offshore seismic lines A-AA-AB" does not add much neither to "implications for thermal energy storage" nor to "imaging of dyke swarms

within the Sorgenfrei Tornquist Zone". It looks detached from the main part of the paper and speculates only on the near-vertical vs steeply dipping dykes which is not of a great importance for the practical aspects of the energy storage site. You refer to Phillips et al. (2017) where they show that Paleozoic dykes can dip at 35° - 50°, so there is no need to explain it here. I do not see value in this section.

Page 21: "if any evidence of dyke emplacement could be found in the Precambrian basement" Permian dykes in Precambrian basement or dykes of any age? What is the possible relevance of their presence or absence to the Permian dykes in the study area? Again, I do not see value in this section.

---

## Referee Comment (RC3) · M. Majdanski (Referee) · 12 Oct 2018

The article presents an interesting case study conected to important problem of the thermal energy storage. The authors use variety of geophysical methods in well thought out analysis that gives an interesting conclusions. Also the study area with dykes directly observed in quarry is a difficult, but interesting case. Gathered data, a combination of wide-angle refractions and reflections, are also good quality and has been collected with state of the art equipment and techniques.

The manuscript is in general written with clear and easy to understand language, at least for not native speaker.

[Figure]

I have seen those results before at the conference, and had a positive impression about the whole concept. However, I got a few comments that, in my opinion, should improve the overall good level of the article.

1) Seismic has been measured with two types of the equipment, cable and wireless system. What was the frequency of the cable system geophones? Was is also 10 Hz as wireless described in text? Why is the noise level so different between the observations with as presented in Fig.5 and 6?

2) In paragraph 3.2 authors describe processing steps. Unfortunately, important prestack data enhancement is not described in details (only mentioned in table 2). What has been used in this step? Also paragraph mention importance of the velocity analysis, but in all processing a constant velocity has been used. Why tomographic results has not been utilised to create a velocity model for further processing steps?

3) Fig.7 - why noisy part of the data is totally muted? It is a critical part of the results. I understand its quality is poor, but at least there should be a hint of the structure.

4) Fig.9 P2 and P3 tomographic results shows very deep and sparse penetration of rays. This might lead to artificial increase of velocities in places marked as B2 and B3, that is further used in the interpretation. This tomographic inversion should be calculated with limited space preventing rays from escaping downwards. Only P4 tomographic results looks realistic.

Some small technical and typographical corrections:

Page 13, line 20 – reference do fig. 12 should be added

Page 14, line1: alone > along

Fig.14 colour scale is missing, isolines are not described

Page 20, line 9: also be dipping -> also to be dipping

Page 21, line 9: Could you please describe what filters has been used?

Page 21, line 17: I see no red dashed lines in Fig.16

---

## Author Comment (AC1) · 13 Nov 2018

(Public reviewer) I found this a very interesting read, partly because the combination of techniques and disciplines is novel to me. I have three comments, one slightly pedantic and two key observations that I think has been overlooked.

(Authors) We thank the public comments of C. Magee. We have addressed all the specific comments below and in our revised manuscript as detailed here.

(Public reviewer) On page 4, Line 27 you state: "Although a matter of a debate, Phillips et al. (2017) claim the first dyke swarm images observed in reflection seismic data in

[Figure]

the world." In Phillips et al (2017) [for transparency I should clarify I'm an author on this paper], it is stated: "While previous seismic-based studies have imaged or contain evidence of one or several dikes, which may or may not be part of a dike swarm (e.g., Zaleski et al., 1997; Malehmir and Bellefleur, 2010; Wall et al., 2010), we present the first seismic data set that images and constrains the geometry of a dike swarm."

(Authors) Thank you for the clarification.

(Authors) We have removed that sentence from the text in our revised manuscript.

(Public reviewer) To clarify our sentence construction here, we agree that other seismic-based studies have clearly identified individual dykes with a dyke swarm (e.g. Wall et al. 2010 image three or four dykes). The orientation of these dykes does provide information on the orientation of the dyke swarm, but it does not constrain other geometrical properties (e.g. width, changes in height) of the dyke swarm. As such, and after an extensive literature search, we think the data presented in Phillips et al. (2017) are the only data that images the entire width of a dyke swarm and does so for >100 km along-strike; thus providing a novel insight into the 3D geometry of a dyke swarm. However, please clarify why you consider this a matter of debate. If we have missed pieces of literature that seismically image the structure of dyke swarms (not just individual intrusions) then we would be very grateful if you could cite them to bring this work to light and make up for any oversight in Phillips et al. (2017).

(Authors) Thank you for the clarification. We read this as if this was the first time dyke-swarms were observed in seismic data and not the whole width of the system as mentioned here. We understood this wrong.

(Authors) We have removed the sentence as mentioned above.

(Public reviewer) A key observation when considering whether dykes intersect storage sites will be spacing between dykes. You clearly highlight that dip and thickness are important. From your images I think you could also say something about spacing.

[Figure]

Bunger et al. (2013) [Bunger, A.P., Menand, T., Cruden, A., Zhang, X. and Halls, H., 2013. Analytical predictions for a natural spacing within dyke swarms. Earth and Planetary Science Letters, 375, pp.270-27] show that dyke spacing should be systematic. The size of the storage site relative to the spacing between dykes could thus be a determining factor in its placement. From all your different data sources, it should be possible to put some quantitative constraints on dyke spacing.

(Authors) Thank you for notifying us on this article. As far as the height goes in the nearby quarry the dolerite dykes are continuing towards depth for over 100 m depth. As far as spacing goes, the dykes are mainly 200-300 m apart but occasionally smaller ones are present coming closer (see Figure 15). Given the history of the Tornquist zone, an estimate while possible it would only be too speculative. Most dykes are judged to have intruded into the earlier faults (extension system) while likely a few opened their ways up. We are a bit puzzled with the article as it refers to "height". If we take the spacing as the known parameters, the height would then be either 2.5 times less (i.e., slightly less than 100 m) or at its deepest point 0.3 times less (i.e., about 1000 m). Would this suggest that mechanically the dykes can extend at depth something around 100-1000 m? This however cannot tell anything about the dip.

(Authors) We therefore do not speculate further on this as the uncertainty is quite huge (10 times). No changes to the manuscript therefor required.

(Public reviewer) There seems to be no mention of what the resolution of the data is. You mention it is 'high-resolution' but there should be some consideration of what this means; in other words, what are you missing? E.g. are <5 m thick dykes imaged and, if not, what could their importance be? There should also be an explanation about tuned reflection packages.

(Authors) With the resolution we both meant fine source and receiver spacing as well as resolving power. Perhaps worth emphasizing that if our interpretation of the reflection from dolerite dykes is correct, the detection limit is somewhere around 10-15 m. This is
not the resolution and rather what the data have been able to image. The detection limit might be better than this but we do not know if there smaller dolerite dykes at where the seismic data have been acquired. On the basis of having useful frequency content between 40-180 Hz and taking 100 Hz as being dominant, we can estimate (using a 4000 ms-1 velocity) a detection limit on the order of a few meters and vertical resolution of 10 m. Judging from the data, it does not appear we have resolved the thickness and rather detected the dykes as one reflection (i.e., top and bottom are imaged as one).

(Authors) We have now added a short text to notify what we mean with high-resolution.

---

## Author Comment (AC2) · 13 Nov 2018

(Anonymous Referee #1) The paper presents results from three high-resolution 2D seismic lines acquired as a first stage characterization of the subsurface for a potential thermal energy storage site in Sweden. The focus of the paper is really on dolerite dykes which are important geological features in the study area. The topic is of interest and relevant to this special issue. In my opinion, the paper should be published after some weaknesses and loose ends are properly taken care of during revision (see main and detailed comments below).

(Anonymous Referee #1) Main comments: 1) The paper either was written quickly or

[Figure]

did not deserve all required attention prior to submission. The language is often imprecise and sometimes sentences are not grammatically correct. This is not systematic throughout the manuscript but prevalent enough to require proper and careful editing by a native English speaker.

(Authors) A native speaker will go through the revised version and we hope this shortcoming is fixed. We have taken additional steps to improve the readability of the text.

(Anonymous Referee #1) 2) The correspondence between log anomalies and reflections (R2 and R3) is particularly good on the un-migrated section (Figure 10b) but rather poor on the migrated section (Figure 10c, especially for R3). What makes you think that the anomalies on logs are really the signature of the dolerite dykes if logs and reflections are not at the same spatial location? Are there any other possible explanations for these log anomalies? The same observation applies to figure 11 (i.e. poor fit with log anomalies and reflection R2 on migrated section). How confident are you about the migration velocities and velocities used for T-Z conversion?

(Authors) We do not have a good control on the migration velocities nor can be 100% sure if the reflections are from the dykes. Based on the relative match and natural gamma/density logs, strength of the reflectivity and how regular they appear in the section we have tried to argue that they are originated from the dykes. Faults have been ruled out because in one of the cases the density increases too. Much of the discussion is to support this interpretation. One reason why the match is not perfect can be due to the 2D nature of the profiles. We can aim to push this to match as much as possible by changing the migration, lower velocities, however we avoided this to be clear. We are now discussing this in the revised manuscript.

(Anonymous Referee #1) 3) I suggest looking for references on physical rock properties of dolerite dykes to support the interpreted signature on logging data (especially low natural gamma) and seismic data. Specifically, the acoustic impedance of dolerite dykes and potential contrast with host rocks are not discussed in the paper. However,

acoustic impedance is the only property that can unequivocally confirm the reflective (i.e., P-wave) nature of dolerite dykes. The interpretation of steeply-dipping reflections as dolerite dykes currently lacks that irrefutable argument (even though strong conjectural arguments are provided to support this interpretation).

(Authors) Additional references are added now. • Planke S, Cambray H (1998) Seismic properties of flood 727 basalts from hole 917A downhole data, Southeast 728 Greenland Volcanic Margin. Proc ODP Sci Results 729 152:453–462 730 • Planke S, Rasmussen T, Rey S et al (2005) Seismic 733 characteristics and distribution of volcanic intrusions 734 and hydrothermal vent complexes in the Vøring and 735 Møre basins. In: Geological Society, London, petro736 leum geology conference series. Geological Society of 737 London, pp 833–844

(Anonymous Referee #1) 4) The paper includes a discussion on the implications of results for thermal energy storage without really providing the key characteristics of a good site for such storage. Some details about an optimal thermal storage site should be provided. For instance, what is the size of such a site ("caverns")? Why is this area considered suitable for thermal storage (particular rock types, geomechanical properties, close proximity to city)? It is difficult for readers to assess the implications of results for something that is not properly defined.

(Authors) Additional details about a good site are added in the introduction. The main reason this specific site was chosen was the proximity to the quarry owned partly by the concept developer (Skanska) as a proxy also for geology and accessibility to the construction site (caverns), if tunneling should be done and of course being close to potential consumers (Lund and Malmö).

(Anonymous Referee #1) 5) Results from the Babel seismic lines are of certainly of interest but primarily from dykes are effective water barrier.

(Authors) We made a mistake with the orientation of the BABEL line section and now it might be that it is more interesting as the lower crustal reflectivity seem to orient the

same way as the interpreted dykes seen in the seismic sections. This is corrected now and text adjusted.

(Anonymous Referee #1) Page 2 line26: ". . .and if major bedrock undulations" with ". . .and determine if major bedrock undulations"

(Authors) Corrected.

(Anonymous Referee #1) Page 6 line 1: "where geological structures are favorable" please be more specific about the geological structures.

(Authors) We meant the dyke systems as they are the most dominant features in the area. A short text added.

(Anonymous Referee #1) Page 8 line 2: replace "where" with "were"

(Authors) Followed.

(Anonymous Referee #1) Page 8 line 9: What explains the better results with diversity stack over conventional stack on line 3?

(Authors) Likely because the noise level (wind and from car traffic) was higher when data acquired along this profile. This is added now.

(Anonymous Referee #1) Page 9 line 12: "Migration was not employed for the data along profile 5". Do you mean profile 4? (Authors) Yes. We have corrected this.

(Anonymous Referee #1) Figure 12 shows mostly dipping reflections, not steep but still dipping reflections. The migrated section should also be shown in the paper for a "proper" positioning of reflectors.

(Authors) The quality of the migrated section for this line is not so desirable and we wish to not show it in the manuscript. If the reviewer insists we can of course show it but it would just be distractive for the readers.

(Anonymous Referee #1) Page 10 line 8: replace "for where needed" with "where

needed"

(Authors) Followed.

(Anonymous Referee #1) Page 11 line 2: "A smooth 2D velocity model". Please provide more information about this model (lower and upper velocities and their distribution (ie. constant gradient?)).

(Authors) Followed.

(Anonymous Referee #1) Page 11 line 14: "suggesting that the tomographic velocity models can be used to constrain the bedrock surface with a good level of confidence". What velocity is the threshold to determine the bedrock surface? In addition, I suggest drawing the top of bedrock on Figure 9 (with dashed line or other).

(Authors) We were unclear on this. Given that there are smoothing constraint used for the inversion, bedrock is not resolved accurately. In our case, we used a rapid change of velocity (vertical gradient) from 3000 m/s to at least 4000 m/s as the bedrock. A dashed line is introduced to represent the interpreted bedrock level in Figure 9.

(Anonymous Referee #1) Page 11 line 23: ". . .depression-looking bedrock is clearly observed in the tomograms suggesting a possibility for major weakness zones (likely highly fractured and/or weathered) in the bedrock." There are other supporting evidences for this elsewhere in the manuscript. I suggest including them here (for example, caption of figure 9 mentions cuttings from bedrock with alteration). What about velocities from tomography – can they help? Also, could you please be more specific about the alteration observed in cuttings?

(Authors) Followed.

(Anonymous Referee #1) Page 13 line 15: "using transverse type approach". What do you mean specifically? (Authors) It meant to be just 2D profiles. Changed to "profiling".

(Anonymous Referee #1) Page 13 line 17: "Several northeast-dipping, approximately

60-65 degree, reflections were imaged down to 400 m depth thanks to the close shot and receiver spacing strategy of the data acquisition." Not sure I understand what is meant by "thanks to close shot and receiver spacing". Aren't large offset required to image steeply dipping reflectors? (Authors) Both required. Spatial sampling and long offsets. Text modified.

(Anonymous Referee #1) Page 13 line 19: "but at occasions are discontinuous and have different appearances". Please be specific about the different appearances.

(Authors) Followed.

(Anonymous Referee #1) Page 13 line 25 "One particular reflection is associated with the topographic depression in the study area and matches well with the high-velocity zone observed under the depression." This is somewhat contrary to what was said before on bedrock depressions corresponding to faults and/or alteration which would normally have low velocity. Please clarify this statement or earlier statement about depression.

(Authors) It is correct. We meant R3 in this case, which is not as undulating as R4 or where HB5 was drilled.

(Anonymous Referee #1) Page 13 line 29: "suggest different geological structures than those observed along profiles 2 and 3". What would be the different structures and why can't they be observed on the two other profiles? Explanations that follow in the 4 lines after are rather vague.

(Authors) We think the central part of P4 is down faulted. The short reflectivity partly means imaging the dykes partly along this profile but also strong faulting that is partly reflected in the magnetic map (see below, the area between HB1 and HB2). We did not show this map before but show it here only.

(Anonymous Referee #1) Page 14 line 10: "..they are nearly equally observed in the country rock and likely can only act as a medium for other geological features.." Not

clear at all what you mean. . .

(Authors) We have rephrased the text. We meant volumetrically equal.

(Anonymous Referee #1) Page 14 line 11: "with some adjustments and better matching. . .". What adjustments and better matching? Please clarify? See also main comment #2. (Authors) We have rephrased the text. See also above. This was addressed also earlier.

(Anonymous Referee #1) Page 14 line 18: ". . .connected to a reduced natural gamma zone (Fig. 11c), likely a dolerite dyke." Why would dolerite dykes have low natural gamma? Any reference to support this?

(Authors) Typically dolerite (a basalt in composition) have very little radioactive elements like K40. We have already worked on this at other sites and provide a reference. See Malehmir and Bellefleur (2010. Ore Geology Reviews)

(Anonymous Referee #1) Page 19 line 8: is "appealing" the right word?

(Authors) Changed to "interesting".

(Anonymous Referee #1) Page 19 line 8: "a pattern for the groundwater flow towards SE (Fig. 14b) and an orientation consistent with the directions of the dykes and major structures in the area (Fig. 1a). Reflections (R1-R4) greatly match this orientation". I would argue that contour lines presented on figure 14b are more complex than described in the statement above which appears a bit oversimplified. Please provide a more precise description and location (i.e. between R3 and R4 near P2 and P3?). Also, why are lines of water table continuous across reflection R1, R2, R3, and R4 if these reflections are water barriers?

(Authors) You are correct. It might be that the down faulting system we referred to in the picture appended here makes the water flow more complicated in the region where R3 and R4 located. It is consistent to be on the same orientation as P2. We have revised the text accordingly. Thanks for noting this valuable information.

(Anonymous Referee #1) Page 21 line 21: ". . . transparent solidified, seismically homogeneous upper mantle" Could you please explain the meaning of solidified in terms of seismic characteristics of the crust?

(Authors) We just think the region there has no long fabric structure for to produce reflectivity. This is what is referred to. We keep the text as it is.

(Anonymous Referee #1) Page 21 line 23: "While there are a number of SW-dipping reflections in the lower-middle crust projecting towards the location of the Sorgenfrei Tornquist Zone, we can not be sure if these are dykes or frozen magma chambers where dykes intruded to the upper crust within the Tornquist zone." Agree. In fact, there are many possible explanations for those reflections. It might be worthwhile to add a few rather than just providing dyke-related possibilities (i.e., dykes and magma chamber that fed dykes).

(Authors) Actually, we made a wrong placement of the orientation on this section. The correct dip is NE, however we do not want to speculate further. We slightly modified the text.

(Anonymous Referee #1) Page 22 line 9: "Firstly, they are not sub-vertical everywhere within the Tornquist zone and they quickly turn steeply dipping in the subsurface as also observed at a few quarries in the area (e.g., Fig. 15. . .". Could you please clearly indicate what you are referring to in Figure 15 (may be use double arrows – one at the top of cliff and the other at the base to point to the feature you are referring to on Fig. 15)?

(Authors) Followed.

(Anonymous Referee #1) Figure 1. Please add coordinates to a) and b). (Authors) Followed.

(Anonymous Referee #1) Figure 2. It would be useful to add viewing direction for a) and b) or general orientation of the pictures. (Authors) The view directions were not

registered unfortunately. We can do it but it will not be very accurate.

(Anonymous Referee #1) Figure 13: The 3D perspective view is very difficult to visualize on this figure (especially for a). I suggest adding axis (x,y,z) that would improve the perspective view. (Authors) Followed.

(Anonymous Referee #1) Figure 14 please add a) and b).

(Authors) Followed.
* * *
[Figure]

[Figure]

**Fig. 1.**

---

## Author Comment (AC3) · 13 Nov 2018

Anonymous Referee #2 The paper nicely illustrates the ability of the seismic method to contribute to the geological evaluation of the thermal energy storage site. The acquisition and processing methodic, parameters and the workflow described clearly in the text, although the actual delineation of the bedrock surface from the tomography is vague. Also, section 6.3 concerning the entire crust structure seems a bit detached from the main part and objectives of the paper. I would recommend the work for publication with some corrections.

(Authors) Please see also responses to reviewer #1. We made a mistake with the

orientation of the BABEL line. We wish to keep this, as it is major structure crossing the zone. The revised manuscript we hope have made this more connected.

(Anonymous Referee #2) Here are my comments and suggestions: Page 2: "The seismic survey had an initial objective of identifying depth to bedrock and if major bedrock undulations could be related to zones of weaknesses (fractured or/and altered) in bedrock." I did not see the comparison of the depth to the bedrock from seismic (diving wave tomography) to drilling results in the paper. You provided some of it in Fig. 9 but did not summarize the results and did not provide any errors bars or discrepancies. Can you clarify this part please?

(Authors) The interpreted bedrock depth and comparison with the boreholes are shown in the revised figure. Uncertainty is not a big issue although we could try to estimate this as we have done in our earlier results using match of the bedrock depth with the velocities observed in the tomograms and various scenarios. Here, only visual interpretation of the depressions is important, therefore no need really to expand this further.

(Anonymous Referee #2) Page 7, line 10: "that the wireless recorders and shooting on the northern side of the road 11 was necessary to enable their imaging" Shooting – yes, but recording did not provide much looking at the shot records in Figs 5 and 6. All the north-dipping reflections in the records are present in the cable part. Wireless part did not contribute much due to short offsets, unfavorable north dip and maybe due to greater noise as indicated in the stacked sections in Figs 7 and 11. To justify this statement you can show some shot gathers from the source locations in the southern ends of profiles 2 and 3. The wireless part did record very good first arrivals, though.

(Authors) This is a good point and we partly agree with this. Shooting was likely important, however with no recording no good shot statics can be obtained and this can complicate imaging due to bad static solution. We avoid showing more figures. We have already 17 figures. This would just distract the readers.
(Anonymous Referee #2) Page 9: "we used a constant velocity of 4000 ms-1 for both migration and time-to-depth conversion" and later "5000 ms-1 should have been used instead" Why 4000m/s velocity was used? According to Fig 9, tomography velocities are closer to 5000m/s at the bedrock (and deeper down) and might be even greater for amphibolite, gneiss and dolerite.

(Authors) Given the target depths are only 500 m or so, the top 10-50 m low velocity zones should also be taken into account. The average velocity was calculated from the tomograms to be on the order of 4000 m/s also our judgment of the first breaks. We accept that we have no control on this, hence also possibly some mismatch of the reflections with the log data.

(Anonymous Referee #2) Page 9, line 12: meant profile 4, not profile 5?

(Authors) Thanks for spotting this. Corrected.

(Anonymous Referee #2) Figure 8: Please indicate what do the colors mean?

(Authors) They represent data density. Added to the caption now.

(Anonymous Referee #2) Page 14: These include (1). . . (2) they have. . . (4) gneissic-amphibolitic (3) is missing

(Authors) Thanks for spotting this. Fixed.

(Anonymous Referee #2) Figure 13: The relations of the features in the 3D view are unclear and impossible to interpret – looks like a 2D image. Perhaps it would be better to plot X-Y-Z axes and connect reflection points by wired surfaces.

(Authors) Followed also as suggested by reviewer #1.

(Anonymous Referee #2) Section "6.3 Revisiting BABEL offshore seismic lines A-AA-AB" does not add much neither to "implications for thermal energy storage" nor to "imaging of dyke swarms within the Sorgenfrei Tornquist Zone". It looks detached from the main part of the paper and speculates only on the near-vertical vs steeply dipping

dykes which is not of a great importance for the practical aspects of the energy storage site. You refer to Phillips et al. (2017) where they show that Paleozoic dykes can dip at 35_ - 50_, so there is no need to explain it here. I do not see value in this section.

(Authors) Please see the revised manuscript. We insist to keep this as the BABEL lines A-AA-AB provide a larger picture crossing the Tornqust zone. We also made a mistake with the orientation of the profile and now further speculations can be made.

(Anonymous Referee #2) Page 21: "if any evidence of dyke emplacement could be found in the Precambrian basement" Permian dykes in Precambrian basement or dykes of any age? What is the possible relevance of their presence or absence to the Permian dykes in the study area? Again, I do not see value in this section.

(Authors) The value would be a more confidence on their orientation and how deep they extend. Please see above and response to Reviewer #1.

---

## Author Comment (AC4) · 13 Nov 2018

M. Majdanski (Referee #3) The article presents an interesting case study conected to important problem of the thermal energy storage. The authors use variety of geophysical methods in well thought out analysis that gives an interesting conclusions. Also the study area with dykes directly observed in quarry is a difficult, but interesting case. Gathered data, a combination of wide-angle refractions and reflections, are also good quality and has been collected with state of the art equipment and techniques. The manuscript is in general written with clear and easy to understand language, at least for not native speaker. I have seen those results before at the conference, and had a

positive impression about the whole concept. However, I got a few comments that, in my opinion, should improve the overall good level of the article.

(Referee #3) 1) Seismic has been measured with two types of the equipment, cable and wireless system. What was the frequency of the cable system geophones? Was is also 10 Hz as wireless described in text? Why is the noise level so different between the observations with as presented in Fig.5 and 6?

(Authors) Both systems used 10 Hz. They were acquired in different days. Traffic and wind noise was stronger.

(Referee #3) 2) In paragraph 3.2 authors describe processing steps. Unfortunately, important prestack data enhancement is not described in details (only mentioned in table 2). What has been used in this step? Also paragraph mention importance of the velocity analysis, but in all processing a constant velocity has been used. Why tomographic results has not been utilised to create a velocity model for further processing steps?

(Authors) For the NMO correction, a variable velocity was used to honor the steeply dipping reflections. This was not a constant velocity. Future studies could benefit from using the tomography data. Here, we kept the processing conventional as already good images are obtained.

(Referee #3) 3) Fig.7 - why noisy part of the data is totally muted? It is a critical part of the results. I understand its quality is poor, but at least there should be a hint of the structure.

(Authors) It is quite noisy and make the main reflection quite weak. We can add another figure but this make the article unfocused. We wish to keep it as it is.

(Referee #3) 4) Fig.9 P2 and P3 tomographic results shows very deep and sparse penetration of rays. This might lead to artificial increase of velocities in places marked as B2 and B3, that is further used in the interpretation. This tomographic inversion should be calculated with limited space preventing rays from escaping downwards.

Only P4 tomographic results looks realistic.

(Authors) This is a good point and we agree. We do not have any critical interpretation below the bedrock level. Rays have channelled as spotted by the reviewer due to likely starting model but also 3D nature of the tomography.

(Referee #3) Some small technical and typographical corrections:

(Authors) Page 13, line 20 – reference do fig. 12 should be added

(Referee #3) Page 14, line1: alone > along

(Authors) Followed.

(Referee #3) Fig.14 colour scale is missing, isolines are not described

(Authors) Figure has been reworked with numbers showing water table label at different locations. .

(Referee #3) Page 20, line 9: also be dipping -> also to be dipping

(Authors) Followed.

(Referee #3) Page 21, line 9: Could you please describe what filters has been used?

(Authors) Followed.

(Referee #3) Page 21, line 17: I see no red dashed lines in Fig.16

(Authors) It is in Figure 16a in the middle of the seismic image.

---

## Author Response (AR2)

**Responses to the reviewers' comments:**

**Seismic imaging of dyke swarms within the Sorgenfrei Tornquist Zone (Sweden) and implications for thermal energy storage**

Alireza Malehmir[1,*], Bo Bergman[2,3], Benjamin Andersson[4], Robert Sturk[4] and Mattis Johansson[2,3]

[1] Dept. of Earth Sciences, Uppsala University, Sweden
[2] Sweco Environment AB, Malmö, Sweden
[3] WSP (presently), Malmö, Sweden
[4] Skanska Sverige AB, Malmö, Sweden

*Correspondence to*: Alireza Malehmir (alireza.malehmir@geo.uu.se)

**Editor comment:**

Comments to the Author:
Dear Authors,
Based on the TE's report and overall assessment of the review process, I am pleased to accept your manuscript for publication on Solid Earth, pending the minor amendments as requested by the TE. Many thanks for choosing Solid Earth.

Sincerely,
Federico Rossetti

(Authors) We thank the editors and referees for finding our revised manuscript acceptable for publication. The very minor comments have also been addressed in the final submission.

**Referee #1:**
The authors have properly addressed my concerns and comments in this revised version of the manuscript. I have only the few minor comments (see below). I do not need nor wish to review the paper again for those changes.
Page 9 line 16: "Migration was not employed for the data along profile 4 because of the gentle dip and this should be noted when the data are interpreted. Unmigrated seismic sections are shown for data quality control purposes and the reflections observed there do not represent actual locations of the geological features nor are dips true". This should be updated since Fig. 12 now includes a migrated version of line 4 (thanks for including this – migrated results are useful and better than described initially).
(Authors) We thank the reviewer for spotting this. We have rewritten this sentence now.

Figure 9: Please add the meaning of the dashed line in the caption.
(Authors) Described now in the caption.

Page line 15: I suggest replacing "This was motivated to check…" with " The motivation for this was to check…"
(Authors) Followed.